# Live-cell imaging of glucose-induced metabolic coupling of β and α cell metabolism in health and type 2 diabetes

Zhongying Wang[1,2], Tatyana Gurlo[2], Aleksey V. Matveyenko[3], David Elashoff[4], Peiyu Wang[1,5,6], Madeline Rosenberger[2], Jason A. Junge[1,5], Raymond C. Stevens [1✉], Kate L. White[1], Scott E. Fraser [1,5,6✉] & Peter C. Butler [2✉]

Type 2 diabetes is characterized by β and α cell dysfunction. We used phasor-FLIM (Fluorescence Lifetime Imaging Microscopy) to monitor oxidative phosphorylation and glycolysis in living islet cells before and after glucose stimulation. In healthy cells, glucose enhanced oxidative phosphorylation in β cells and suppressed oxidative phosphorylation in α cells. In Type 2 diabetes, glucose increased glycolysis in β cells, and only partially suppressed oxidative phosphorylation in α cells. FLIM uncovers key perturbations in glucose induced metabolism in living islet cells and provides a sensitive tool for drug discovery in diabetes.

[1] Department of Biological Sciences, Bridge Institute, Michelson Center for Convergent Bioscience, University of Southern California, Los Angeles, CA, USA. [2] Larry L. Hillblom Islet Research Center, David Geffen School of Medicine, University of California, Los Angeles, Los Angeles, CA, USA. [3] Department of Medicine, Division of Endocrinology, Metabolism, Diabetes, and Nutrition, Mayo Clinic School of Medicine, Rochester, MN, USA. [4] Department of Biostatistics, University of California, Los Angeles David Geffen School of Medicine, Los Angeles, CA, USA. [5] Translational Imaging Center, University of Southern California, Los Angeles, CA, USA. [6] Biomedical Engineering, University of Southern California, Los Angeles, CA, USA. ✉email: stevens@usc.edu; sfraser@usc.edu; pbutler@mednet.ucla.edu

In health, blood glucose levels are maintained in a narrow range to provide adequate glucose for the brain, and to avoid the adverse consequences of hyperglycemia. This is accomplished by the regulated secretion of the pancreatic islet hormones insulin and glucagon in response to the prevailing glucose concentration. In diabetes, by definition, this regulation is defective. β cell stress and dysfunction precede progressive loss of β cell mass in both type 1 diabetes (T1D) and type 2 diabetes (T2D). In T1D, β cell stress is mediated by autoimmunity[1]. The cause of β cell stress and dysfunction in T2D is not fully resolved, but protein misfolding and formation of toxic oligomers of islet amyloid polypeptide (IAPP) are emerging as important contributors[2]. In both T1D and T2D, β cell dysfunction is at least in part due to remodeled glucose metabolism as a component of a conserved pro-survival signaling program[3,4]. In healthy β cells, as glucose concentrations increase, there is a proportionate increase in adenosine triphosphate (ATP) generation from oxidative phosphorylation (OxPhos) that, in turn, acts on membrane $K_{ATP}$ channels, electrically coupling oxidative phosphorylation to insulin secretion[5]. In stressed β cells, it has been proposed that there is increased glycolysis that is no longer coupled to oxidative phosphorylation in both T1D and T2D, consistent with the so-called pro-survival Warburg metabolism prominent in cancer cells[3,4].

To date, most studies of β cell metabolism have relied on strategies such as metabolomics and/or gene array data to gain an aggregate impression of the sum of cells within islets. Single cell transcriptome studies of such markers in islets are inconsistent and are limited by signal strength[6]. The role of glucose metabolism in the regulation of glucagon secretion by α cells in health, or dysregulated secretion in diabetes, is unclear[7].

In prior live cell imaging studies of pancreatic β and α cells, Piston's group used two photon excitation microscopy to measure the glucose induced NADH and NADPH [in sum, hereafter referred to as NAD(P)H] autofluorescence in islet cells, and documented a greater heterogeneity of responses in dispersed cells compared to cells within islets[8]. Using the same approach in flattened single cells adherent to matrix proteins, they noted the majority of the glucose-induced NAD(P)H increase corresponded to the mitochondrial network[9]. The phasor approach of fluorescence lifetime imaging microscopy (FLIM) offers a powerful means to further exploit the autofluorescence of NAD(P)H to deduce cellular metabolism, and has been used to establish the trajectory of metabolism during stem cell differentiation[10], cancer development[11–13] and in multiple disease models[14–16]. The fluorescence lifetime of free NAD(P)H is 0.4 ns, but increases to ~3 ns when bound to proteins such as Complex I in the electron transfer chain[17,18]. NADH generated by glycolysis[19], and NADPH by the pentose phosphate pathway, are mainly unbound to proteins; and thus can be recognized by their shorter lifetimes[20]. In contrast, NADH generated by the tricarboxylic cycle is recognizable by the longer lifetime resulting from its binding to proteins in the electron transport chain, predominantly Complex I[19]. This characteristic difference in the lifetimes of NAD(P)H autofluorescence permits the phasor-FLIM technique to offer an analytical tool to monitor shifts in glycolysis and oxidative phosphorylation over time in individual living cells, even within live tissue[14,21,22].

Since glucose intolerance is an early abnormality in both T1D and T2D, our overall goal was to establish the trajectory of metabolism in both healthy and stressed β cells and α cells from the same islets, in response to an increase in glucose from basal to glucose stimulated levels. We validated the phasor-FLIM technique in INS-1E cells, and then in pilot studies applied it to dispersed primary mouse β and α cells under basal and increased glucose levels to establish application of phasor-FLIM in primary islet cells. By combining FLIM and confocal imaging we were able to extend these insights, and establish β and α cell responses to increased glucose concentration in intact islets where cell-cell interactions persist, capturing the heterogeneity of cell responses over time. To determine cellular metabolic trajectory in the setting of β cell stress, we used a well-characterized mouse model of T2D, based on IAPP toxic oligomer induced β cell stress characteristic of T2D in humans[23]. Finally, we evaluated human islets from organ donors with and without T2D.

## Results

**FLIM quantification of oxidative phosphorylation in live pancreatic β cells.** In order to validate the use of phasor-FLIM to quantify oxidative phosphorylation in living β cells, we first applied FLIM imaging in the pancreatic β cell line INS-1E (Fig. 1). These cells are easily grown as a monolayer and are relatively flat, simplifying the evaluation of FLIM signals from distinct subcellular compartments (Fig. 1a). We detected the autofluorescence generated by NAD(P)H in the 440nm-500nm emission range after 2-photon excitation at 740 nm, presented as an intensity map (Fig. 1b). The lifetime of this autofluorescence was analyzed using the phasor approach, permitting the fluorescent lifetimes recorded from each pixel to be used to establish the metabolic state of each region: short lifetime NAD(P)H fluorescence corresponds to free NAD(P)H generated by glycolysis; longer lifetime NAD(P)H fluorescence reflects the bound state generated by oxidative phosphorylation (Fig. 1c). The autofluorescence lifetime distribution of NAD(P)H in the phasor map is pseudocolored to present the relative proportions of free versus bound NAD(P)H in each pixel: blue depicts a predominance of short-lived fluorescence (predominantly glycolysis); green, yellow and red represents progressively greater fractions of long-lived fluorescence (predominantly oxidative phosphorylation) (Fig. 1c).

INS-1E cells were evaluated under conditions of low and high glucose (Fig. 1d, e), and demonstrated the changes in OxPhos expected in the pseudocolored images.

The metabolic state measured by FLIM is presented as Bound/total NAD(P)H between 0 and 1. This Bound/total NAD(P)H is calculated from the projected position of the median along the metabolic state axis, obtained by first calculating the median of the points in the histogram and then projecting the median onto the linear regression fit to the phasor-FLIM data. A high Bound/total NAD(P)H value reveals a metabolic state with high oxidative phosphorylation while a low Bound/total NAD(P)H indicates predominant glycolysis[22]. The metabolic trajectory revealed by phasor-FLIM is as anticipated, with the Bound/total NAD(P)H in INS-1E cells moving from a relatively low value at low glucose concentrations to an increased value on glucose stimulation (Fig. 1f, g).

To validate that the Bound/total NAD(P)H value obtained from phasor-FLIM reflects the relative amount of OxPhos and glycolysis, we analyzed INS-1E cells after inhibition of glycolysis with 2-deoxyglucose (Supplementary Fig. 1a). As expected, this inhibition resulted in an increase in the ratio of Bound/total NAD(P)H. In contrast, uncoupling oxidative phosphorylation from the electron transport chain by application of potassium cyanide resulted in the expected fall in Bound/total NAD(P)H (Supplementary Fig. 1a).

The subcellular regions of high Bound/total NAD(P)H (Fig. 1d, e) have the appearance of the mitochondrial network, consistent with NADH bound to the protein complexes of the electron transport chain. To test this assumption, we undertook FLIM followed by detection of the mitochondrial network by immunofluorescence of Tom20 in INS1-E cells. The filamentous

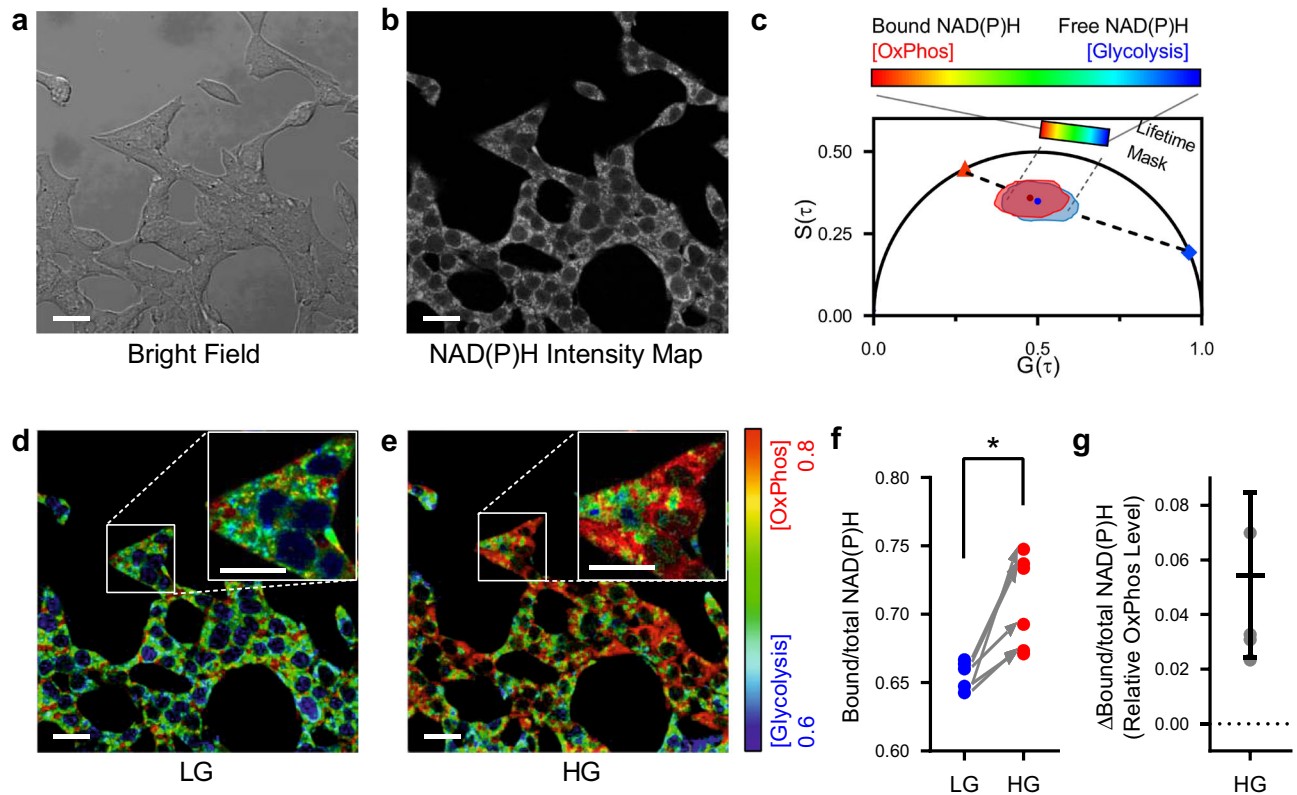

**Fig. 1 Validation of FLIM in a pancreatic β cell line.** INS-1E β cells were monitored by FLIM at low glucose (LG) concentration 1.1 mM and then after increasing the glucose concentration to 16.7 mM (high glucose, HG), a value known to stimulate insulin secretion in this cell line. The typical INS-1E adherent and flat morphology is seen in bright field (**a**) permitting a clear view of intracellular compartments. **b** 2-photon excitation of the same field yields a fluorescent intensity map gated for NAD(P)H emission. **c** The phasor plot of the fluorescence lifetime of the NAD(P)H signal offers a simple way to analyze and visualize FLIM data, as the emission from NAD(P)H will lie along the dashed line: the red triangle (lifetime = 3.43 ns) marks the phasor position of NAD(P)H bound to protein complexes, and the blue diamond (lifetime = 0.4 ns) marks the phasor position of free NAD(P)H, unbound to proteins. The position along this dashed line can be rendered by the depicted rainbow color code: red represents the longer lifetimes, corresponding to the high Bound/total NAD(P)H indicating oxidative phosphorylation (OxPhos); blue represents the shorter lifetimes corresponding to glycolysis. **d, e** The fluorescence lifetimes from NAD(P)H recorded in low glucose (LG) and high glucose (HG) conditions show the expected metabolic state, with the pseudocolored lifetimes of most cells moving to redder hues. **f** To analyze this metabolic state, the average Bound/total NAD(P)H value is plotted, revealing the expected increase as conditions are changed from LG to HG, reproducibly in 6 experiments. **g** The relative metabolic state between conditions (e.g., LG to HG) can be expressed as ΔBound/total NAD(P)H, which is positive for a relative increase in OxPhos, and negative for a relative increase glycolysis. Data are presented as mean ± SD from n = 6 independent experiments. Scale bar, 10 μm. *P < 0.05, two-tailed Wilcoxon test.

pattern and distribution of the highest Bound/total NAD(P)H signal coincides with the mitochondrial network (Supplementary Fig. 1b). However, the mitochondria are not the only source of signal; for example, when pyruvate is metabolized by pyruvate decarboxylase, NADPH can be generated in mitochondria or cytoplasm by isocitrate dehydrogenase[24].

**FLIM measurement of oxidative phosphorylation in primary islet cells reveal differential responses of β and α cells to glucose.** Having validated phasor-FLIM to evaluate the relative extent of glycolysis and oxidative phosphorylation in INS-1E cells, we extended its use to dispersed primary mouse islet cells from wild type (WT) non-diabetic islets (Fig. 2). By imaging the cells adherent to slides printed with a grid, we were able to perform FLIM evaluation of the live cells and subsequently identify the two main islet endocrine cell types, β and α cells, after fixation and immunocytochemistry for insulin and glucagon (Fig. 2a, b). We quantified the NAD(P)H lifetime distribution for 8 β cells and 8 α cells, in response to basal glucose (BG; 4 mM) and high glucose (HG; 16 mM) concentrations (Fig. 2c, Supplementary Fig. 2).

At basal glucose concentrations there was a clear difference in metabolic state between β and α cells. β cells had a relatively low signal for oxidative phosphorylation confined to the perinuclear region; in contrast, α cells displayed a pronounced oxidative phosphorylation signal [higher Bound/total NAD(P)H] in a filamentous pattern evenly distributed through the cell. As expected, there was an increase in oxidative phosphorylation in β cells in the high glucose condition; this increase in Bound/total NAD(P)H present in a filamentous pattern distributed through the cell. In contrast, α cells showed the opposite metabolic trajectory, reacting to high glucose with a decrease in Bound/total NAD(P)H, and redistributing the high OxPhos signal to the peripheral submembrane segments of the cell. The discordant metabolic trajectories of α cells and β cells are illustrated by plotting Bound/total NAD(P)H values and ΔBound/total NAD(P)H (Fig. 2d, e, f).

**Evaluation of live cell glucose metabolism in intact mouse islets.** The functional properties of pancreatic endocrine cells are impacted by paracrine cross talk between cells within the islet. Therefore, we used phasor-FLIM to evaluate the relative amounts of glycolysis and oxidative phosphorylation in intact living mouse

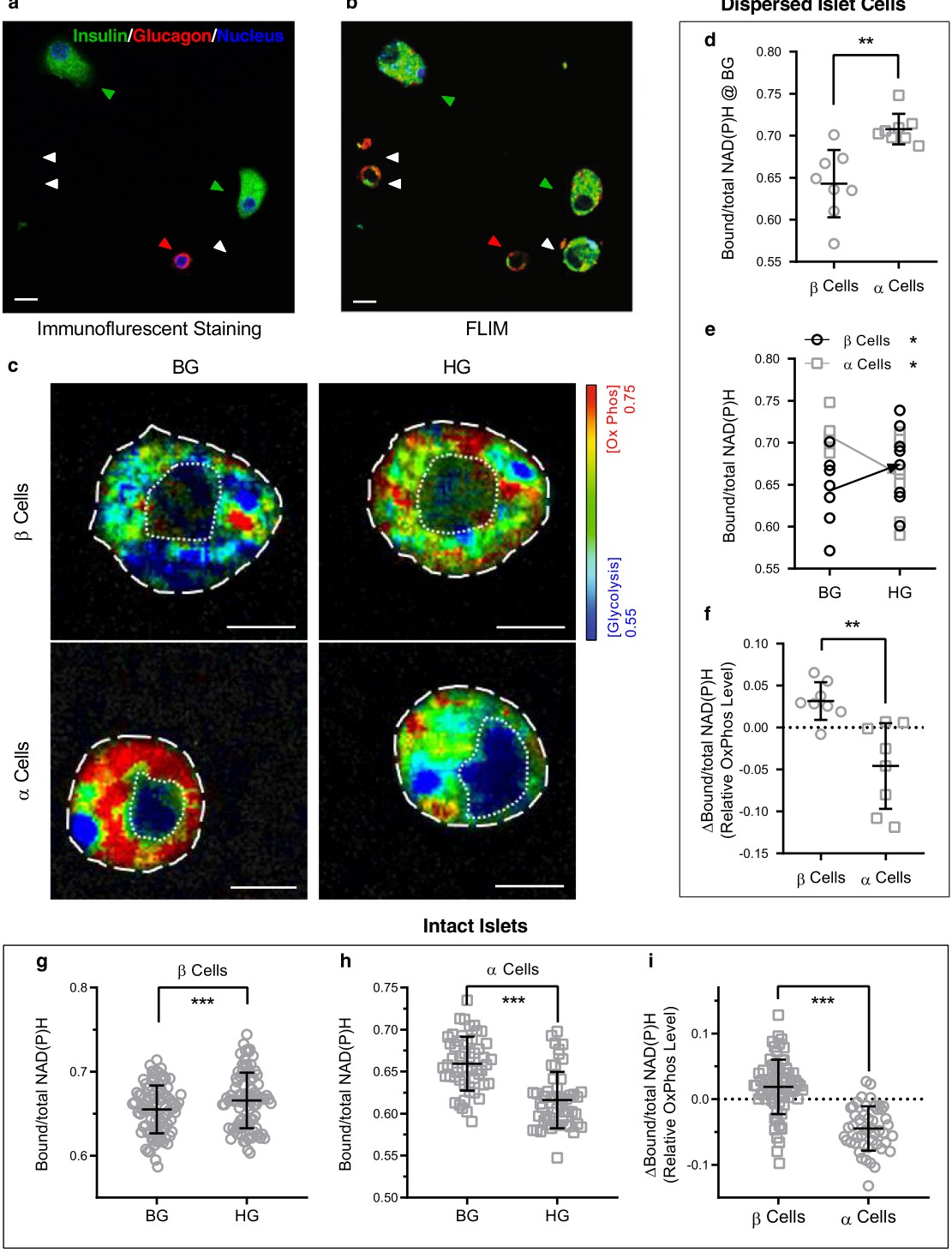

**Fig. 2 Metabolic trajectory of β and α cells in response to glucose. a, b** Mouse islets were dispersed into single cells, attached to a cover slip with imprinted grid, evaluated by FLIM and then fixed and immunostained for glucagon and insulin (**a**) so that the FLIM studies could be attributed to α cells and β cells, respectively. Green arrows, β cells; red arrows, α cells; white arrows, unknown because cells detached during staining. Scale bar, 10 μm. **c** Metabolic trajectory of dispersed Single primary β and α cells (*n* = 8 cells each) imaged at basal glucose (BG, 4 mM) and after exposure to high glucose (HG, 16 mM) for 30 min. The pseudocolored images suggest that the β cells become more OxPhos after glucose treatment and the α cells become more glycolytic. Scale bar 5 μm; dashed white line indicates cell outline and dotted line indicates the nucleus outline. **d** The basal glucose Bound/total NAD(P)H values for α and β cells are significantly different. **e, f** The opposite metabolic trajectories of dispersed α cells and β cells are revealed by plotting the Bound/total NAD(P)H values for each cell in BG and HG conditions, and calculating the ΔBound/total NAD(P)H of β and α cells. *n* = 8 for each cell type. **g, h, i** In 8 intact WT mouse islets, we examined the Bound/total NAD(P)H and ΔBound/total NAD(P)H value of 47 α and 72 β individual cells at basal and high glucose. Data are presented as mean ± SD. \**P* < 0.05, \*\**P* < 0.01, the two-tailed paired *t* test was used for comparison of responses to BG versus HG in dispersed mouse islet cells. The two-tailed Mann–Whitney test was used in comparison of responses between α versus β cells from dispersed mouse islet cells. Linear mixed effect models were used to compare responses between islets cells in whole islets.

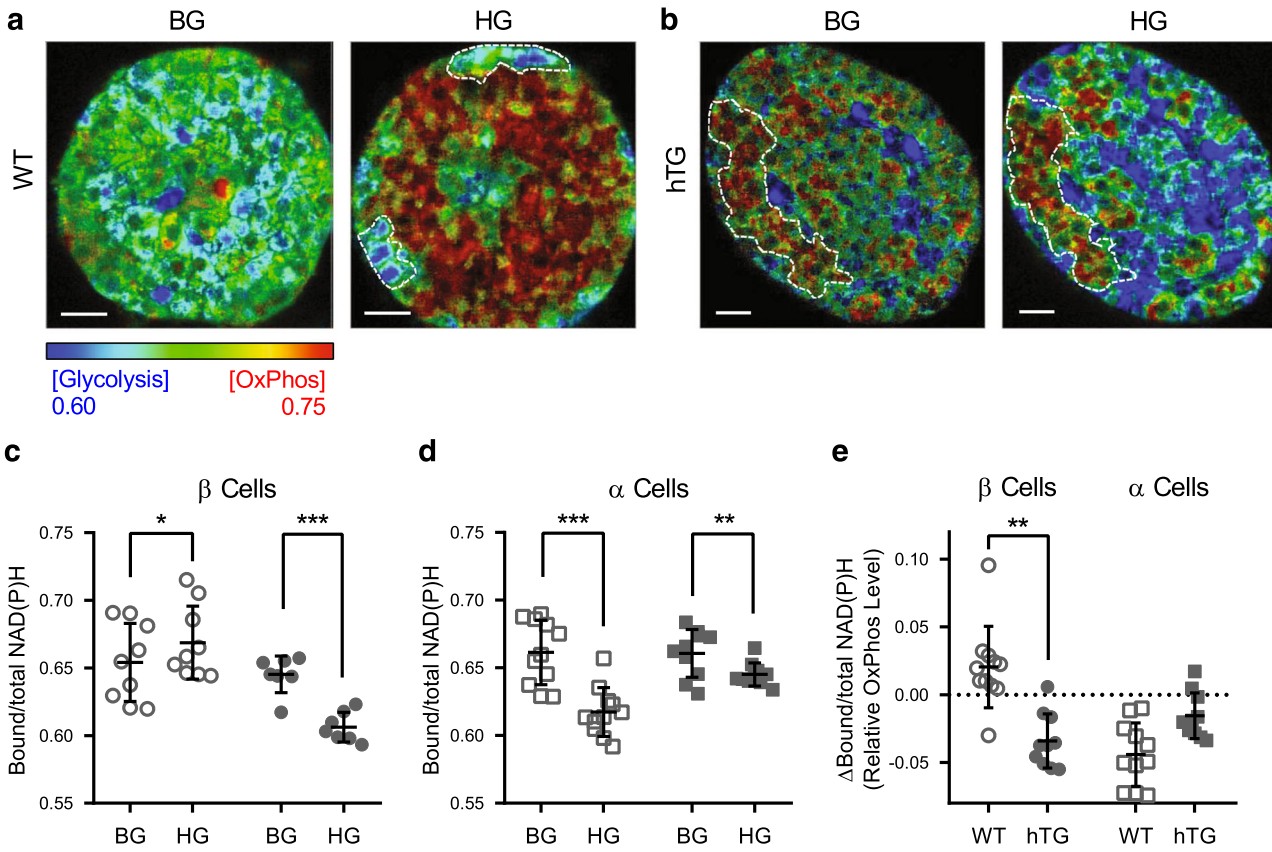

**Fig. 3 Impact of protein misfolding stress characteristic of T2D on metabolic trajectory of β and α cells in response to glucose.** FLIM images depicting relative glycolysis and OxPhos in islets from a wild type (WT), (**a**) and human IAPP transgenic (hTG), (**b**) mouse islet at basal glucose (BG, 4 mM) and high glucose (HG, 16 mM). **c, d** Panels show Bound/total NAD(P)H value for β cells and α cells respectively in WT (open circle and square) and hTG (solid circle and square) mouse islets at BG and HG. **e** Relative increase (+) or decrease (−) in OxPhos (ΔBound/total NAD(P)H) in α and β cells from WT and hTG mouse islets after increase in glucose from BG to HG. Data are presented as mean ± SD, $n = 11$ for WT islets, 1–3 islets from 6 mice, $n = 10$ for hTG islets, 1–3 islets from 4 hTG mice. *$P < 0.05$, **$P < 0.01$, ***$P < 0.001$; the linear mixed effect models were used for statistical analysis. Scale bar, 10 μm.

islets in response to physiological basal and stimulated glucose concentrations. In mouse islets[25], α cells represent only ~10% of total islet cell numbers but are usually located in the periphery of islets; in contrast, the far more abundant β cells occupy the core of the islet. Therefore, the aggregate phasor-FLIM signature for an intact islet will be dominated by the contribution of β cells, but the optical sectioning ability of two-photon and confocal laser scanning microscopy permits the selective imaging of the α cells positioned in the periphery. Cell identity can be verified by fixation and immunostaining of islets after FLIM imaging.

Analysis of the metabolic trajectory in individual α and β cells within 9 intact islets from WT mice at basal (4 mM) and then high glucose (16 mM) concentrations (Fig. 2g, h and i) revealed changes in FLIM comparable to those observed in dispersed α and β cells. The Bound/total NAD(P)H increased in β cells within intact islets in response to high glucose ($P < 0.001$), implying increased oxidative phosphorylation; in contrast, the Bound/total NAD(P)H decreased in α cells ($P < 0.001$). The net WT mouse islet response to an increase in glucose concentration from basal (4 mM) glucose concentrations to high glucose (16 mM) showed the β cells in the core of the islet moving from low levels of oxidative phosphorylation (with a few cells with higher levels of oxidative phosphorylation scattered within the islet) to a marked increase in oxidative phosphorylation. This results in the overall transition from green to red in the color-coded NAD(P)H bound/total images (Fig. 3a). In contrast, phasor-FLIM showed that the α cells (enclosed by white broken lines; Fig. 3a) in WT mouse

islets had higher oxidative phosphorylation at basal glucose levels that decreased in response to exposure to high glucose (color red/green to blue), consistent with the response observed in dissociated α cells (Fig. 2d).

**Disordered β and α cell glucose metabolism in a mouse model of type 2 diabetes.** T2D is characterized by misfolded protein stress induced by formation of toxic membrane permeant oligomers of IAPP, a protein co-expressed and secreted with insulin by β cells[2]. Islet dysfunction in T2D is characterized by both impaired insulin secretion and failure to suppress glucagon secretion in response to hyperglycemia[26]. In order to investigate the β and α cell responses to increased glucose concentrations under conditions of IAPP-induced toxicity, we compared islets from mice transgenic for human IAPP (hTG) with islets from WT mice (Fig. 3a, b). The islets analyzed were procured from hTG mice at age 9 weeks, before diabetes onset to avoid the confounding actions of hyperglycemia. The distributions of α and β cells within the islets are less stereotyped in hTG islets, but areas of α cells, corroborated by later immunostaining, were selected for FLIM analysis (broken white lines in Fig. 3b and Supplementary Fig. 3). In hTG islets, the majority of β cells showed a predominant increase in glycolysis in response to high glucose, in stark contrast to the increased oxidative phosphorylation observed in WT β cells (Fig. 3c). Such a remodeled metabolism had previously been inferred from metabolome and enzyme expression studies of islets from humans with T2D and islets

from hTG mice[4]. This inferred inversion of the normal response of β cells to glucose stimulation is confirmed directly here in living hTG islets. Oxidative phosphorylation was comparable in α cells in hTG and WT islets at basal glucose, and decreased in response to high glucose in both WT and hTG islets. However, the suppression of α cell oxidative phosphorylation was diminished in hTG islets in comparison to WT islets (Fig. 3d, e). In WT α cells, mitochondria with an active oxidative phosphorylation signal were mostly confined to the cortical region of the cells, adjacent to the cell membrane; but in hTG mice the signal was more distributed throughout the cytoplasm at high glucose.

Collectively these data support the premise that glucose metabolism contributes to regulation of glucagon secretion, and that dysregulated glucagon secretion in T2D might be a consequence in part of remodeled glucose metabolism. Moreover, these data in mice with β cell specific IAPP toxicity further support the premise that failure to suppress glucagon secretion in response to hyperglycemia in T2D is secondary to β cell dysfunction rather than a primary α cell defect.

**Recruitment of β cells to active oxidative phosphorylation by high glucose.** One of the strengths of FLIM is the ability to monitor the metabolic state of living cells over time. It has been proposed that time to activation of β cells in response to an increase in glucose concentration is heterogeneous, with some β cells being activated early, and then apparently acting as a hub to recruit adjacent β cells through a wave of depolarization[27]. By evaluating the metabolic state of the same islets at basal (4 mM) glucose concentrations and then for 120 min after increasing the glucose concentration to 16 mM, we had the opportunity to examine the time course of activation of individual β cells as well as the topological relationship between the cells and their timing of activation. In islets from WT mice, FLIM confirmed the marked heterogeneity between β cells in the time to activation in response to an increment in glucose concentrations (Fig. 4a). Moreover, as proposed by the hub β cell model, cells juxtaposed to the first cells to show active oxidative phosphorylation (white circles) were subsequently activated, often resulting in a radiating pattern of activation from the first responders. In contrast, islets from hTG mice showed a progressive increase in glycolysis, not an increase in oxidative phosphorylation, in response to an increment in glucose concentration, and there were no obvious hubs of early responding cells initiating this change.

**Confirmation β and α cell dysfunction in the HIP rat model of T2D.** Having established the metabolic trajectory of β and α cells in response to glucose in the setting of β cell IAPP toxicity characteristic of T2D in humans, we next sought to relate this to insulin and glucagon secretion in vivo, using the most physiologically relevant glucose challenge: oral glucose. Because blood volume limits repeat blood sampling in mice, we performed oral glucose tolerance tests in the human IAPP transgenic (HIP) rat model of T2D[28]. The HIP rat has the same β cell human IAPP transgene that is expressed in hTG mice, and displays comparable β cell misfolded protein stress. HIP rats and WT controls were evaluated at 7 months of age, an age that precedes overt diabetes onset, but recapitulates many features of islet failure in patients with T2D[28]. Consistent with previous studies, HIP rats had glucose intolerance, with both impaired glucose stimulated insulin secretion apparent from the attenuated C-peptide response (~twofold decrease in peak C-peptide in HIP vs. WT, $P < 0.05$) despite hyperglycemia (Fig. 4b, c and d). We have previously shown that impaired glucose-mediated insulin secretion in the HIP rat mimics that in humans with T2D, with loss of insulin pulse mass and decreased hepatic insulin clearance accounting for

the relatively higher post hepatic insulin compared to C-peptide in the HIP rats[28,29]. Glucagon concentrations were comparable in HIP and WT rats at basal glucose but failed to suppress in HIP rats after glucose ingestion, despite the higher glycemic excursion ($P < 0.05$ for % glucagon suppression in HIP vs. WT, Fig. 4e).

These in vivo studies compliment the FLIM studies in isolated living islet cells. They affirm that β cell specific misfolded protein stress present in T2D causes impaired glucose-mediated insulin secretion as well as defective glucose-mediated suppression of glucagon secretion, both characteristics of islet dysfunction in T2D.

**Live cell imaging of altered glucose metabolism in human islets from T2D.** Having undertaken live cell imaging studies to define the coupling of metabolism to a physiological increase in glucose concentration in β and α cells in health and under conditions of β cell misfolded protein stress characteristic of T2D, we next translated these studies to the clinical setting. We employed the phasor-FLIM approach to human islets procured from 3 non-diabetic (ND) and 2 T2D organ donors (appendix table 1). Due to the more dispersed distribution of α cells in human islets compared to mouse islets, we were unable to reliably identify regions of α cells in living human islets[25]. To identify β cells, we took advantage of the readily identifiable lipofuscin bodies characteristic of human β cells to then evaluate the metabolic response of human β cells to an increased in glucose concentration from 4 to 16 mM[30]. Consistent with studies in WT mouse islets, the FLIM signal for β cell oxidative phosphorylation activity showed an apparent increase in response to glucose in the ND islets (Fig. 5). In contrast, consistent with the mouse model of T2D, there was no increase in the FLIM signal for oxidative phosphorylation in β cells in the islets from T2D.

## Discussion

In the present studies, we sought to address the following questions: what is the metabolic trajectory of β and α cells in response to an increase in glucose concentration in health, and how does this differ under conditions of β cell stress present in T2D? By use of FLIM we were able to address these questions directly in living islet cells monitored over time.

The increase in oxidative phosphorylation in healthy β cells in response to an increment in glucose is consistent with prior studies[31]. In healthy β cells engaged in active oxidative phosphorylation, the mitochondrial network is in a highly fused state, extending through the cytoplasm as detected here by FLIM. It has long been proposed that there is heterogeneity between β cells in responsiveness to glucose[32]. It has also been proposed that there are early responders to an increment in glucose that then serve as hubs to recruit surrounding β cells, presumably through electrical coupling[27].

By FLIM monitoring over time we were able to directly document the recruitment of β cells to active oxidative phosphorylation in response to an increment in glucose. The concept of islet hub cells that recruit a progressive wave of surrounding cells was supported by these observations. Given the timing of this process, it is likely that recruitment of responders contributes to the so called second phase of insulin secretion that has generally been assumed to be a consequence of mobilization of stored insulin vesicles versus the release of insulin from the readily releasable docked and primed insulin secretory vesicles[33]. Also of note, the relatively slow time course of this process is consistent with the point recently made that this apparent wave of recruitment is slower than would be expected by electrophysiological coupling, leaving the mechanism of the observation to be fully established[34].

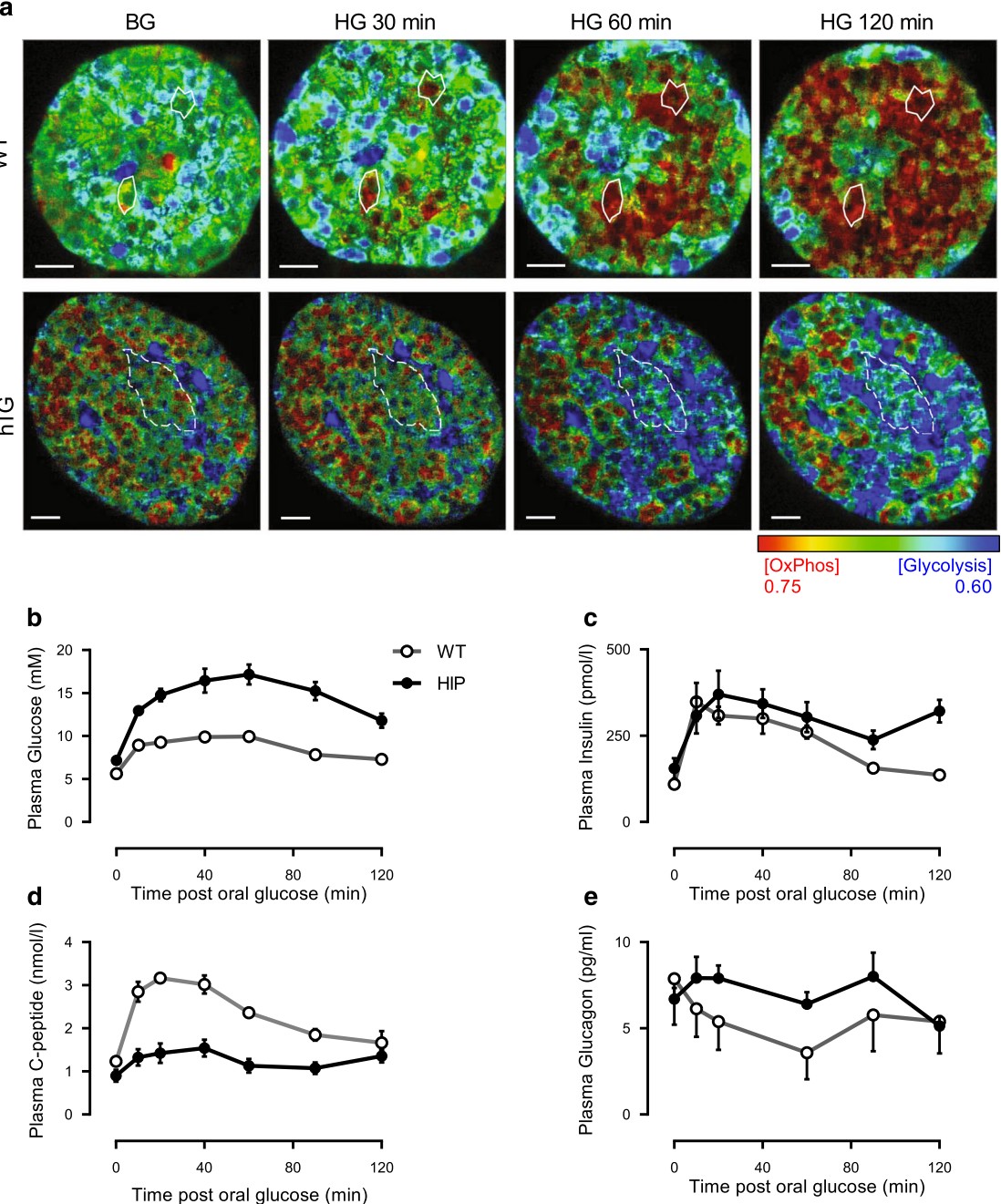

**Fig. 4 Spatial temporal relationship of β cell metabolic responses to glucose in vitro and insulin and glucagon secretion in vivo in the presence and absence of misfolded protein stress. a** FLIM evaluation of metabolic trajectory from representative wild type (WT) and human IAPP transgenic (hTG) mouse islets with basal glucose (BG, 4 mM) and then at 30, 60 and 120 min after high glucose (HG, 16 mM). White solid line, hub β cells observed in WT islets; white dashed line, a cluster of adjacent β cells in hTG mice. Scale bar, 10 μm. Plasma glucose (**b**), insulin (**c**), C-peptide (**d**), and glucagon (**e**) concentrations at baseline (min-0) and following 1.5 g/kg body weight oral glucose by lavage in WT ($n = 5$) and HIP ($n = 11$) rats. Data presented as mean ± SEM.

In contrast to healthy β cells, cells impacted by hIAPP toxic oligomers characteristic of T2D showed increased glycolysis, but not oxidative phosphorylation, in response to an increment in glucose. This is consistent with the accumulating evidence that stressed β cells in T2D and T1D have remodeled glucose metabolism that disengages glycolysis from oxidative phosphorylation in response to injury response signaling[3,4]. This change has been inferred from gene expression and metabolome studies, but is now shown here directly by using FLIM to monitor the metabolic trajectory of living islet cells before and after exposure to an increment in glucose.

There is not only a progressive loss of glucose stimulated insulin secretion, but also loss of pulsatile insulin secretion in models of β cell hIAPP toxicity, reproducing defective insulin secretion in T2D[28,29]. It was therefore of interest that there was no effective hub pacemaker activity in the presence of hIAPP toxicity. This might be because of the impact of hIAPP toxicity on the putative hub cells or loss of cell to cell electrical coupling consequent on disruption of membrane integrity by hIAPP oligomers[35].

Failure to suppress glucagon secretion contributes to hyperglycemia in both T1D and T2D[36]. While this might be attributed at

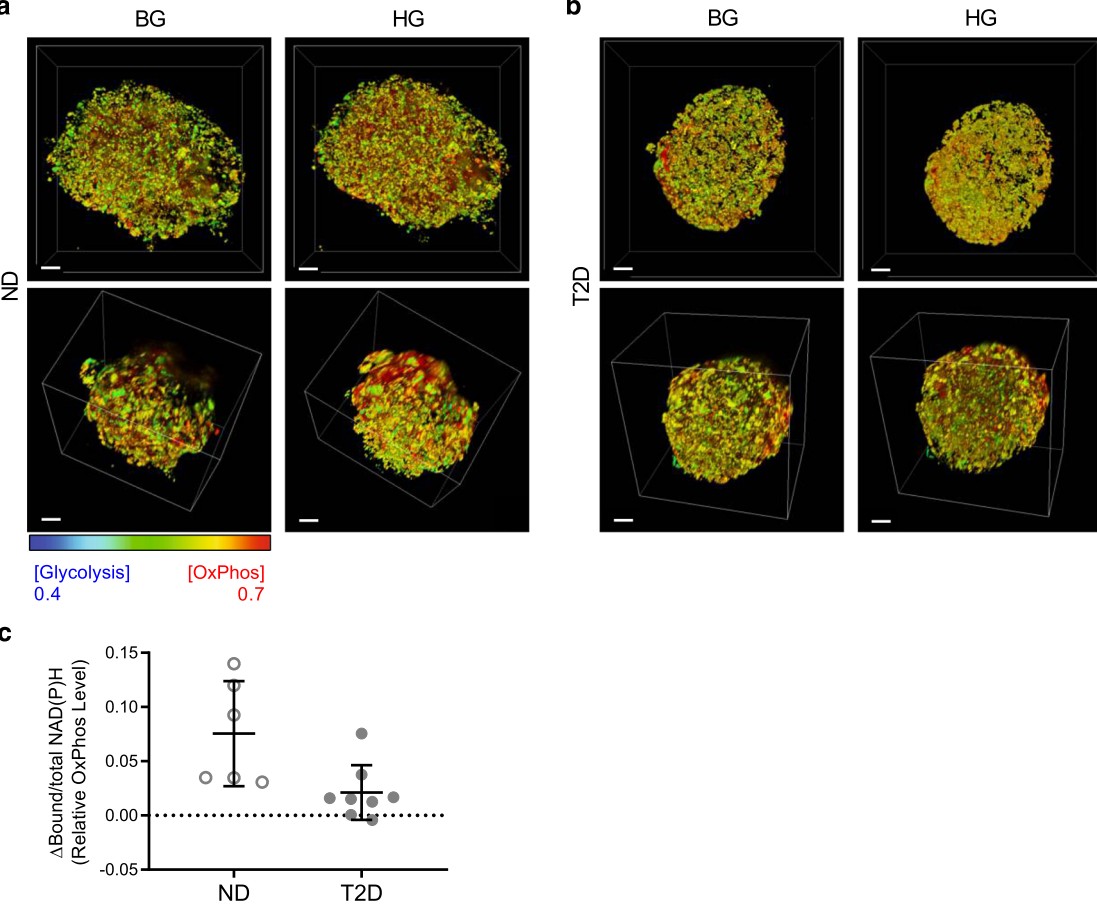

**Fig. 5 Human islets.** 3D projection of human islets from non-diabetic donors (ND) (**a**) and donors with T2D (**b**) exposed to basal glucose (BG, 4 mM) and high glucose (HG, 16 mM) concentration. **c** Quantification of changes in relative OxPhos levels of β cells in ND and T2D human islets presented as mean of each donor sample. $n = 8$ for T2D (2 donors, 4 islets each), $n = 6$ for ND (3 donors, 1-3 islets each). Scale bar, 20 μm. $P = 0.1$, linear mixed models were used for statistical analysis.

least in part to loss of the paracrine suppression by insulin, there is an increasing interest in the intrinsic regulation of glucagon secretion. In common with β cells, the ATP/ADP ratio increases in α cells with increasing glucose concentrations[37]. In the present studies, by use of FLIM we resolved the subcellular site of the generation of ATP by oxidative phosphorylation activity in dispersed α cells in the basal and glucose stimulated state. In the absence of extrinsic factors, there is an apparent change in the subcellular site of oxidative phosphorylation activity in α cells, being present throughout the cell at basal glucose concentrations but confined to the submembrane region at high glucose concentrations. This finding is consistent with total internal reflection fluorescence (TIRF) microscopy measurement of ATP levels in the submembrane region of α cells in response to glucose[38]. Current models for the regulation of glucagon secretion imply that activation and closure of the $K_{ATP}$ channels in α cells at the plasma membrane inhibit glucagon secretion[37]. This is hypothesized to occur due to membrane depolarization following closure of $K_{ATP}$ channels suppressing $Na^+$ channel activity that in turn attenuates the amplitude of action potentials that activate voltage gated P/Q type $Ca^{2+}$ channels required for exocytosis of docked glucagon secretory vesicles. Focusing oxidative phosphorylation at the submembrane site in response to increased glucose appears to be a particularly efficacious adaptation, since it not only delivers ATP at high concentrations directly to the $K_{ATP}$ channels, but the locally active mitochondria presumably clear any cytosolic $Ca^{2+}$ further restricting glucagon exocytosis.

The oxidative phosphorylation distributed throughout the cell at basal glucose values might be mediated by β-oxidation of long chain free fatty acids, availability of which would be suppressed by increased insulin levels at higher glucose[39]. It has been proposed that oxidation of free fatty acids generates ATP that sustain α cell action potentials which in turn increase glucagon secretion. Therefore, the regulation of glucagon secretion might be mediated through the balance between stimulation of glucagon secretion, by ATP generated by free fatty acid oxidation that provides energy to sustain $Na^+$ channel driven action potentials, and the more topographically restricted glucose-mediated oxidative phosphorylation, at the submembrane site, delivered directly to $K_{ATP}$ channels that suppresses glucagon secretion. Future studies will be required to confirm this refined model of regulated glucagon secretion and how it is perturbed in T2D.

In summary, it is well known that metabolism couples hormone secretion by β and α cells to the circulating glucose concentration. Most available data in this regard is inferred from single time points and the sum of cells per islet. Direct imaging of live cell metabolism in pancreatic islets by phasor-FLIM establishes the direction and extent of metabolic transitions in individual islet cells, and even compartments within cells, over time. We established that β cell misfolded protein stress, characteristic of T2D, remodels metabolism in both β and α cells, disrupting the usual tight coupling of ambient glucose concentrations with regulated hormone secretion. Phasor-FLIM provides a powerful tool to investigate therapies to restore β and α cell function in diabetes.

## Methods

**Cell culture**. INS-1E cells (kindly provided by Dr. Pierre Maechler, University Medical Center, Geneva, Switzerland) were cultured in RPMI 1640 medium with 11 mM glucose supplemented with 5% heat-inactivated fetal bovine serum, 100 Units/ml penicillin, 100 µg/ml streptomycin, 2 mM glutamine, 10 mM HEPES, 1 mM sodium pyruvate and 50 µM β-mercaptoethanol at 37 °C in a humidified 5% $CO_2$ atmosphere.

INS-1E cells were seeded on Chambered cover glass (VWR, Cat# 62407-056) precoated with Poly-L-lysine (Sigma-Aldrich, Cat#P7280). Coating was performed as follows: Poly-L-lysine was diluted in PBS to 0.1 mg/mL Poly-L-lysine, and applied to the culture surface at 1.0 mL/25 cm² for 5 min at room temperature, followed by rinsing with water and air drying for 2 h inside the biosafety cabinet. Cells were plated at a density of 70,000 cells per cm² and cultured for 3–4 d to reach 70% confluency before imaging.

**Animal models**. Animal studies were performed in compliance with the guidelines of the UCLA Office of Animal Research Oversight; ethical approval was obtained from the Research Safety & Animal Welfare Administration at UCLA (ARC#2004-119 and ARC#2004-114). Wild type mice were originally purchased from Charles River Laboratory, and the colony was maintained at UCLA. The generation of the transgenic mouse expressing human IAPP (FVB-Tg(IAPP)-6Jdm/Tg(IAPP)6Jdm or hTG) was described elsewhere[23]. The wild type rats (CD Sprague Dawley originally from Charles Rivers) and the generation of hemizygous human islet amyloid polypeptide (h-IAPP) transgenic rats (HIP rats) as a validated rodent model of T2D recapitulating human islet pathology[28,40].

**Rat glucose tolerance studies**. A total of 5 Sprague-Dawley rats (WT) and 11 rats expressing human IAPP (HIP rats) at ~7 months of age were used for current study. Rats were bred and housed in pairs under standard environmental conditions with a 12-hr light-dark (LD) cycle and habituated to routine handling, restraint, and oral gavage. After 12 h food deprivation, rats were administered an oral glucose tolerance test (OGTT) by oral gavage of 1.5 g/kg body weight of dextrose and blood was collected via lateral saphenous venipuncture into chilled, EDTA-treated microcentrifuge tubes before (0 min) and 10, 20, 40, 60, 90, and 120 min post oral gavage. Plasma glucose was measured by the glucose oxidase method using the YSI 2300 STAT Plus Glucose & Lactate Analyzer (YSI Life Sciences). Plasma insulin and C-peptide were measured using the ALPCO Diagnostics Rat Insulin ELISA kit (80-INSRT-E01) and Rat C-peptide ELISA kit (80-CPTRT-E01), respectively. Plasma glucagon was measured using the Mercodia Glucagon ELISA kit (10-1271-01).

**Mouse islets and single islet cells**. Islets were isolated from 9 to 10 wk old normoglycemic male mice (Supplementary Table 1) by collagenase digestion method as described in detail[41]. Mouse islets (200–250) were washed once with PBS, pelleted and resuspended in 0.5 ml of ice-cold Accutase cell detachment solution (Innovative Cell Technologies, San Diego, CA, USA, Cat#AT-104) and incubated in water bath at 37 °C for 5 min with gentle trituration after 3 min. Equal volume of tissue culture medium (TCM, RPMI with 11 mM glucose supplemented with 100 IU/ml penicillin/streptomycin and 10% fetal bovine serum) was added to stop reaction. Islet cells were spun for 3 min at 1000 g and resuspended in TCM for plating on Ibidi-slide 8 well with grid (Ibidi USA, Fitchburg, WI, USA, Cat# 80826-G500) precoated with Poly-L-Lysine (Sigma-Aldrich, St. Louis, MO, USA, Cat#P6407) followed by Laminin (Life Technologies, Carlsbad, CA, USA, Cat#23017015) as described elsewhere[42]. Whole islets (80–100 µM diameter) were plated at a density 4–6 per well and imaged 3 d later. Dispersed islet cells were plated at a density 30,000 per well; imaging was performed after 3–5 d in culture.

**Human islets and single islet cells**. Human pancreatic islets were provided by the National Institution of Diabetes and Digestive and Kidney Diseases (NIDDK) funded Integrated Islet Distribution Program (IIDP) at City of Hope, National Institute of Health (NIH) Grant # 2UC4DK098085. The characteristics of islet donors are listed in Supplementary Table 2. Human islets were maintained in suspension in RPMI 1640 medium with 5.5 mM glucose supplemented with 100 IU/ml penicillin/streptomycin and 10% fetal bovine serum at 37 °C in a humidified 5% CO2 atmosphere. Islets were cultured in suspension for 4–6 d after isolation before plating on ibidi-slide 8 well with grid.

Whole islets (80–100 µM diameter) were plated at a density 4–6 per well, imaging was performed 3 days after plating. Slides were precoated with extracellular matrix by growing human bladder cell carcinoma HTB-9 (human bladder cell carcinoma line 5367, available from ATCC) to a confluence and then lysed by 10 min exposure to 20 mM NH4OH.

**Imaging strategy**. For FLIM imaging, a Leica SP8 FALCON with DIVE laser scanning microscope system was used. A water immersion objective 40x/1.1NA was used for islets and dispersed islet cells image acquisition. Dispersed islet cells were imaged in multiple positions with a spiral scanning at the beginning to keep track of cell positions. Oil immersion objective 63x/1.4NA was used for cell line, and immunofluorescent image acquisition. Pinhole was set to 1 AU for fluorescence image acquisition to enhance lateral resolution and reduce out-of-focus

signal in deep tissue. NAD(P)H and lipofuscin FLIM signal was collected using 740 nm 2-photon excitation (Spectra-Physics InSight X3 tunable laser, 0.8 mW average power), 440–500 nm and 560–620 nm emission (Supplementary Fig. 4). Excitation intensity and number of frames collected were adjusted to collect >100 photons/pixel in the cytoplasm; thus, more than 10^5 photon counts contributed to the phasor histograms in each of the optical sections.

INS-1E cells were starved in 1.1 mM glucose KRBH buffer (pH 7.4, 111 mM NaCl, 25 mM NaHCO3, 4.8 mM KCl, 1.2 mM KH2PO4, 1.2 mM MgSO4, 10 mM HEPES, 2.3 mM CaCl2 and 0.1% bovine serum albumin (BSA)) for 30 min in the microscope incubator and then treated with 16.7 mM glucose KRBH buffer for 30 min. FLIM imaging was taken right before and 30 min after glucose stimulation. Z-step size is 1 µm. FLIM data were collected in 512 × 512 pixel format at 400 Hz.

Dispersed islet cells or intact islets were initially maintained in KRBH buffer with a glucose concentration of 4 mM for 1 h in the microscope incubator to reach temperature and pH balance. We evaluated the islets by FLIM prior to addition of glucose to media to reach a concentration of 16 mM and then again evaluated the islets by FLIM at 30, 60, and 120 min. The Z-step distance varied from 1.5 to 2 µm, depending on islet size. Islet NAD(P)H FLIM data were collected in 256 × 256 pixel format at 0.4 frame per sec.

Relative location of dispersed islet cells was recorded by tile scanning prior to experiment and after immunostaining to verify cell types. Identified cells were subjected to phasor analysis.

**Data analysis strategy**. Phasor FLIM distributions (G,S coordinates) are derived from a Fourier transform[43] of lifetime data. Each pixel in the image corresponds to a point in the phasor plot. Desirable z planes were selected and a 5 × 5 median filter was applied by LAS X FLIM/FCS (Version 3.5.6, Leica Microsystems, Inc, Buffalo Grove, IL). FLIM G and S coordinates data were further processed and analyzed using customized threshold setting (Supplementary Table 3) and manual region of interest (ROI) in Matlab (Version R2019a, MathWorks, Inc, Natick, MA). For both rodent and human islets, an upper threshold was applied to remove pixels with extreme high lipofuscin signal and a lower threshold was applied to remove pixels of background and nuclear signal (Supplementary Fig. 4, Table 3). For each data set, the Bound/total NAD(P)H value was determined using a linear regression line fit to the G and S for each cell in the set, with the fixed point of 100% free NAD(P)H (tau 1). The other intersection of the regression line with the universal circle of the phasor plot (tau 2) represents the mean lifetime of 100% Bound NAD(P)H for that data set. The Bound/total NAD(P)H for each data point was calculated by projecting each GS value onto the regression line, and then related to the distance between tau 1 and tau 2. The mean of the Bound/total NAD(P)H ratio is presented in Supplementary Table 4. ΔBound/total NAD(P)H is defined as the Bound/total NAD(P)H after treatment minus Bound/total NAD(P)H before treatment.

To analyze single cells in the intact islets, 4–6 α cells and 8–9 β cells with clear boundary and correct relative position were selected individually from a single Z layer. The region of interest (ROI) of each cell was extracted to calculate Bound/total NAD(P)H and ΔBound/total NAD(P)H. To analyze intact islets, regions of α or β cell were selected and extracted to calculate populational Bound/total NAD(P)H and ΔBound/total NAD(P)H. The Z planes in islets were chosen to be between 10 and 30 µm from the attached bottom of the islets to get segmented distribution of different cell types.

**Immunofluorescence**. The cells or islets were fixed in 4% paraformaldehyde at room temperature for 20 min, washed 3x with PBS. Slides were soaked in 0.4% TritonX-100 for 30 min and blocked with 3% BSA/ 0.2% Triton X-100. The guinea pig anti-insulin (Abcam, Cambridge, MA, USA; Cat#195956, 1:400) and mouse anti-glucagon (Sigma-Aldrich, St. Louis, MO, USA; Clone K79bB10, Cat#G2654, 1:1000) antibodies diluted in TBS supplemented with 3% BSA and 0.2% Tween 20 were applied as cocktail overnight at 4 °C, followed by the cocktail of corresponding conjugated secondary antibodies. Secondary donkey anti-guinea pig and anti-mouse antibodies were F(ab)2 fragments conjugated to FITC or Cy3 respectively from The Jackson Laboratories (West Grove, PA, USA; Cat# 706-096-148 and Cat#715-165-151), and used at dilution of 1:200. Slides were mounted with Prolong Antifade Glass with NucBlue (Thermo Fisher Scientific, Cat# P36981).

**Statistics and reproducibility**. Data are presented as mean ± SD or mean ± SEM from n independent experiments as indicated in the figure legends. Statistical analysis in dispersed WT mouse islet cells was performed using the Mann–Whitney test for comparison of FLIM signals between α and β cells. The paired Wilcoxon signed rank test was used to compare FLIM responses to high versus low glucose in INS1E cells and two-tailed paired t test in dispersed WT mouse islet cells (Graphpad Prism, Graphpad Software, Inc, San Diego, CA).

In order to compare the FLIM signals between groups of interest in intact rodent and human islets (e.g., α vs β or basal vs high glucose) we constructed linear mixed effects models with fixed effects for a group and random effects to account for repeated observations within the same experimental units (ex: random mouse or humans effects and/or random islet within mouse or human). These models were run using the mixed procedure in SAS V9.4 (SAS Institute, Cary, NC). A value of $P < 0.05$ was considered statistically significant. 8 dispersed α cells were analyzed from 2 WT mice and 8 dispersed β cells were analyzed from 1 WT mouse.

11 whole islets were analyzed from 6 WT mice and 10 whole islets from 4 hTG mice (1-3 islets/mouse). Mice characteristics are listed as Supplementary Table 1.

**Reporting summary**. Further information on research design is available in the Nature Research Reporting Summary linked to this article.

## Data availability

The data underlying the findings of this study are available within the Supplementary Information files, Supplementary Data 1, or are available upon requests to the authors.

## Code availability

All codes used for data analysis are available upon requests to the authors.

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

## Acknowledgements

These studies were supported by the United States Public Health Services National Institute of Health grants (DK059579 to P.C.B. and DK098468 to A.V.M.), the NIH National Center for Advancing Translational Science (NCATS) UCLA CTSI Grant Number UL1TR001881 (to D.E.), the Larry L. Hillblom Foundation (2014-D-001-NET) (to P.C.B.), Translational Imaging Center, Leica Microsystems, the Bridge Institute at the USC Michelson Center for Convergent Bioscience, and the USC Office of the Provost. We acknowledge the technical assistance of Tristan Grogan, Department of Medicine Biostatistics Core at UCLA for help with data analysis. We would like to thank Patrick Rorsman University of Oxford and Jakob Knudsen University of Copenhagen and the

members of the Pancreatic Beta Cell Consortium at USC for their inspiring discussions and feedback. We acknowledge the excellent editorial assistance of Jessica Garcia.

## Author contributions

Z.W. and T.G. designed and performed experiments, analyzed data and wrote the manuscript with input from all authors. A.V.M. designed and performed HIP and WT rat experiments. P.W. designed and coded imaging analysis programs. M.R. performed experiments. J.A.J. provided technical support on hardware and imaging acquisition. D.E. performed the statistical analysis. R.C.S., K.L.W., and L.L.W. provided critical input. P.C.B. and S.E.F. designed the studies, analyzed data, and made final edits.

## Competing interests

The authors declare no competing interests.
