## [Peer Review File · Communications Biology]

Reviewers' comments:

Reviewer #1 (Remarks to the Author):

The study by Wang et al. addressed a critical need in the application of metabolic imaging, using NAD(P)H fluorescence of β and α cells for detection and diagnostics of diabetes. The concept of FLIM using to monitor metabolism of β and α -cell is straight forward, the results are convincing and the manuscript is written in a clear and consistent way. Thus it, definitely, can be made suitable for publication.

However, I would like to ask some questions:

How long human and mouse islets were maintained in suspension before the experiment. Will the indicated changes persist in the conditions in vivo?

How many mice did you take in the experiment? Did you track cells from each mouse or combine them? Could you please explain.

It may be worthwhile to present a tables with typical fluorescence lifetime of NADH for different experiments.

Reviewer #2 (Remarks to the Author):

This manuscript presents findings on monitoring islet cell metabolism through phasor-FLIM analysis of cellular autofluorescence. Below are specific comments:

1) Starting with David Piston's work, there have been previous studies over the last 15 years that have used NAD(P)H autofluorescence to monitor islet cell metabolism. These previous studies are not cited, and it is important for the authors to explain how their work fits within the previous literature in this area.

2) The authors report a change in the G-coordinate of phasor, which is a bit interesting. Could they explain why they chose this as a metric? They could have included both G and S coordinates in finding the magnitude of a vector. Their metabolic trajectory in Fig. 1C shows how NAD(P)H does not simply follow along the G axis. Furthermore, the G coordinate of unbound and bound NAD(P)H can be determined and does not range from 1 to 0. Other groups have extrapolated linear curve fits to determine bound fractions from phasor plots. It is also worth noting in the manuscript that phasor space doesn't scale linearly, so a change in G in one location on the map may be quite different from a change in another location.

3) How are upper and lower bounds for color maps determined? The color bars in all figures for example do not have numbers, but they should. The fact that there are two color bars shown in Fig. 1c adds to the confusion.

4) Previous groups have identified lipofuscin as a potential confounding factor in NAD(P)H measurements and sought to remove the lipofuscin contribution. If the authors indicate the presence of lipofuscin bodies in β cells, how did they ensure that they did not affect their metabolic measurements?

5) How were the "desirable" z planes selected and how were images of samples registered before and after treatment? It would also be helpful to show immunostaining that guided segmentation.

6) There are some potential concerns about statistical analysis. Assessments involving 16 cells seems somewhat low. How many independent biological replicates did this come from? Have the authors considered donor-donor or culture-culture variability? T-tests may not be appropriate.

7) The discussion is somewhat speculative. The manuscript would be strengthened if there were supporting measurements of ADP/ATP or membrane potential for example.

8) The authors reference the mitochondrial network multiple times, but some of their autofluorescence signal also is coming from NAD(P)H in the cytosol. This is important to consider when discussing intracellular heterogeneity and the qualitative differences they observe. Is it possible that some differences may not be related to the mitochondria?

9) Minor:

a. Why is G described as a function of tau in the plots? This is unusual.

b. Is nucleus included in all measurements? Has very different lifetime.

c. The authors write "NADH generated by oxidative phosphorylation is bound to protein complexes and has a longer lifetime". This does not make sense because OX PHOS generally generates NAD⁺, not NADH. The bound state is not "generated" by OX PHOS as they indicate in another location in the manuscript. The bound state is generated when NADH binds to a protein complex

,and this is not necessarily specific to Complex I.

d. The authors mention confocal microscopy in a couple locations. If they actually performed confocal microscopy (involving use of a confocal pinhole) and did not just use the laser scanning capabilities of the SP8 for TPEF, please clarify why and what the pinhole was set to.

e. There is a typo in the statistics section ($p < 0.5$).

f. What was the integration time for FLIM?

Reviewer #3 (Remarks to the Author):

Pancreatic Islets and Autofluorescence based α - β separation have piqued interest in diabetic research for nearly four decades (Van de Winkel et al. 1982). Plenty of new information on islet-cell oscillations, calcium stimulus-response, and other studies have made critical advances with the help of autofluorescence-based imaging.

In this work, Wang et al. describe an autofluorescence based Glucose-FLIM assay to study metabolic responses in the pancreatic cell line (INS-1E). The authors demonstrate glucose sensitivity of the NAD(P)H FLIM method for islet cells and the ability to distinguish α and β cells using the relative change in the g -coordinate of the phasor plot. The method is further used to study the islets of transgenic diabetic mouse model (IAPP) and human islets from diabetic and non-diabetic donors. Although the work does a wonderful job in concisely present a multi-model imaging scheme for pancreatic research, I find the technical part lacks clarity in many instances.

My comments are listed below.

* It is not clear which experiments were complemented with confocal/staining to identify α / β cells. If it was used only for dispersed single cells, how were the mouse-islets separated?

* Authors mention the "metabolic trajectory"; but never shows a 2D phasor plot with the metabolic trajectory. For phasor analysis experts, it would be reassuring to demonstrate the phasor trajectory as well. Only a representative image was shown in figure 1. For this representative plot, the figure does not address how the endpoints of the color-trajectory were derived. Was this experimental?

* Line 22: Abstract: Authors mention that the work explores "glucose metabolism with regulated hormone secretion." This is misleading, in my opinion. Consider changing this to more appropriate results.

* Line 22: Abstract: "In healthy cells, glucose enhanced": consider changing "glucose"-> "addition of glucose or Stimulus of glucose or concentration of glucose or similar."

* The notation $G(\tau)$ for frequency domain / Fourier coefficients is confusing. Commonly used/accepted terms are $g(\omega)$ or g . $G(\tau)$ is conventionally used for lag time characteristics like FCS.

* Why was the "primary islet cells reveal differential response" study done for cell culture instead of islets? Islet staining/confocal could have been a more convincing case for the following section on islet cell separation based on morphology and FLIM (section: "glucose metabolism in intact mouse islets"). Are the dispersed islets challenging to handle on an ibidi-marked petri-dish? Other than experimental difficulties, is there another reason to not evaluate on islets instead of single cells?

* Line 105: "In contrast, lifetime signals indicate a predominance of glycolysis in the cell nucleus at both low and high glucose." This is hard to read from the figure panels 1d,1e. Without either a quantification of the nucleus signals or a reasonable link to the manuscript, it would be better not to comment on metabolism in the nucleus in the results section. This seems out of the scope of this work.

* Fig 1 Panel f): Please provide a representative phasor in Figure 1 for LG-HG condition. This is a critical plot to validate phasors.

* I am not convinced by the use of ΔG as a "relative oxphos level." This scale is not necessarily linear for FLIM(lifetimes) or metabolism and can create systematic errors in interpretations. I have seen a similar approach in another work by Kumar et al, 2020: <https://doi.org/10.1128/mBio.02730-18>. However, a direct correlation of relative respiration in cells and ΔG is still to be proven with a complementary technique. Please supplement the use with published reports or flux experiments.

* Line 110: "The $G(\tau)$ score is the median lifetime vector (phasor plot center) of detected fluorescence as a fraction of 1: 1 representing fluorescence that decays quickly; and values near 0 representing very long lifetime (0 = background light).": What do authors mean by "median lifetime vector"? Is there a median calculation in the frequency domain used here? Was the kernel implemented in the phasor-space or time-space? I am also facing difficulty in understanding 1:1 relation of $\tau=0$ and $\tau=\infty$. Please explain what the scoring used for the parameter $G(\tau)$.

* Figure 1.f: How many images were measured per experiment, and was a cellular segmentation used, or was an image averaged? As seen in panel 1E, not all cells are equally responding to the stimulus.

* Figure 2 Panel E: please plot the two cells separately without overlap. The clutter makes it hard to see the changes to individual cells.

* Line 141: "there was an increase in oxidative phosphorylation in β cells with high glucose, extending through the mitochondrial network": Do the figures prove this claim of "extending through the mitochondrial network"? I see an increased frequency of local peaks, not a network response.

* Line 208: "However, the extent of the mitochondrial network that showed suppression of oxidative phosphorylation was greater in WT than hTG α cells ($p < 0.01$):" I find the mention of mitochondrial network speculative. The figure nor the analysis does not measure the mitochondrial network extent.

* Line 433: "100 photon/pixel" What was the accumulated photon count per pixel? Are 100 photons enough for separating cells using the $G(\tau)$ parameter? Was this previously studied/reported?

* Line 438: "FLIM data were collected in 512 x 512 pixel format at 400 Hz." : Was a 2.5ms pixel dwell time used for imaging? Is that a safe practice with 2-Photon excitation? In this context, what was the laser power used?

* Line 441: "FLIM was then undertaken prior to addition of glucose" -> consider replacing the verb undertaken for FLIM.

* Figure 5: Could authors use a color scale proportional to ΔG used to show the 3D volumes? From the presented study, it seems like ΔG is more sensitive than the phasor trajectory. In that case, a more appropriate color scale amplifying the change in g -score could increase the contrast between the samples presented in panels a and b.

Apart from the comments listed above, I find the article exciting because of the vast amount of specimens examined and the raw effort put towards correlating metabolism using a developing science of autofluorescence-diagnostics. I appreciate the authors effort and wish the best of luck with the submission.

Referee expertise:

Referee #1: FLIM and metabolism

Referee #2: FLIM and metabolism

Referee #3: FLIM

Reviewers' comments:

Reviewer #1 (Remarks to the Author):

The study by Wang et al. addressed a critical need in the application of metabolic imaging, using NAD(P)H fluorescence of β and α cells for detection and diagnostics of diabetes. The concept of FLIM using to monitor metabolism of β and α -cell is straight forward, the results are convincing and the manuscript is written in a clear and consistent way. Thus it, definitely, can be made suitable for publication.

Response. Thank you for your encouraging and positive comments.

However, I would like to ask some questions:

(1). How long human and mouse islets were maintained in suspension before the experiment. Will the indicated changes persist in the conditions in vivo?

Response. Thank you for this good point.

The mouse islets were not maintained in suspension before experiments but put on PDL and laminin coated slides immediately after isolation and imaged within 3-5 days. Human islets were maintained in suspension for 4-6 days before plating.

To really address the interesting question by the reviewer, one would have to perform FLIM on islets in situ in the pancreas in vivo and then again after isolation. The best we can offer is that the findings in the isolated islets correlate well with hormone secretion in vivo.

For example, the marked increase in measured oxidative phosphorylation in WT beta cells in response to an increase in glucose corresponds to glucose induced insulin secretion and the relative loss of increased oxidative phosphorylation in hIAPP expressing islets corresponds to the relative loss of glucose induced insulin secretion in these islets in vivo (Figure 4D). This reproduces the pattern of attenuated glucose induced insulin secretion observed in isolated islets from human IAPP transgenic mice compared to wild type mice shown below where islets were cultured in the same chamber slides used for FLIM imaging and at the same basal and stimulated glucose concentrations.

(2). How many mice did you take in the experiment? Did you track cells from each mouse or combine them? Could you please explain.

Response. Good point. We have revised the methods section to include the number of mice studied from each group and for each experiment, numbers of islets or dispersed cells per experiment.

(3). It may be worthwhile to present tables with typical fluorescence lifetime of NADH for different experiments.

Response. Thank you for this excellent suggestion. As requested we now add a table with the mean bound NADH ratio for each experimental condition in Supplementary Table 4.

Reviewer #2 (Remarks to the Author):

This manuscript presents findings on monitoring islet cell metabolism through phasor-FLIM analysis of cellular autofluorescence. Below are specific comments:

Response. We thank the reviewer for the detailed and helpful critique.

1) Starting with David Piston's work, there have been previous studies over the last 15 years that have used NAD(P)H autofluorescence to monitor islet cell metabolism. These previous studies are not cited, and it is important for the authors to explain how their work fits within the previous literature in this area.

Response. Good point and we apologize for this omission. This has now been addressed in the introduction.

2) The authors report a change in the G-coordinate of phasor, which is a bit interesting. Could they explain why they chose this as a metric? They could have included both G and S coordinates in finding the magnitude of a vector. Their metabolic trajectory in Fig. 1C shows how NAD(P)H does not simply follow along the G axis. Furthermore, the G coordinate of unbound and bound NAD(P)H can be determined and does not range from 1 to 0. Other groups have extrapolated linear curve fits to determine bound fractions from phasor plots. It is also worth noting in the manuscript that phasor space doesn't scale linearly, so a change in G in one location on the map may be quite different from a change in another location.

Response. In order to address this suggestion, we reevaluated the computational approach as proposed by the reviewer. We agree that the phasor space does not scale linearly and have made this point as suggested. We have now used both G and S coordinates to calculate the magnitude of the vector as suggested. The metabolic trajectory is now established by using linear regression analysis of all median G and S values. We then map each data point to the regression lines to calculate the bound NADH percentage in each data set. On the regression line, the bound NADH percentage change scales linearly, equaling the ratio of the distance between data point and Free NADH point to the distance between data point and Bound NADH point. Thank you for this helpful suggestion.

3) How are upper and lower bounds for color maps determined? The color bars in all figures for example do not have numbers, but they should. The fact that there are two color bars shown in Fig. 1c adds to the confusion.

Response. The upper and lower bounds for color maps are determined by the phasor distribution of all panels within the same figure. The phasor plot of each figure is overlaid as illustrated in Figure 1C. The upper and lower bounds fall onto the edges of the phasor peak to best illustrate the metabolic change. The red-blue color bar on the top shows the larger view of the Lifetime mask, which paints the corresponding pixels of figures based on the phasor plot.

We have now added the ratio of bound NADH based on the-phasor distribution (Supplementary Table 4). We also now provide the original phasor plot of each figure in the Supplementary Figure 2C for

reference. The yellow-blue color bar on the left indicates the pixel counts of the phasor cloud. As requested we have remade Figure 1C to better illustrate the determination of color map scales.

4) Previous groups have identified lipofuscin as a potential confounding factor in NAD(P)H measurements and sought to remove the lipofuscin contribution. If the authors indicate the presence of lipofuscin bodies in β cells, how did they ensure that they did not affect their metabolic measurements?

Response. Thank you for raising this point. Yes, we did address the lipofuscin signal and we apologize for not being sufficiently clear in the methods on this, now rectified. Lipofuscin within human islets increases with age (Trillian Gregg, Diabetes 2016) but is much less abundant in the young mice included in the present study.

Lipofuscin was readily detectable in beta cells in humans islets, and as previously stated, was used to identify beta cells. Of note, there is no difference in lipofuscin in beta cells in T2D compared to nondiabetic controls (Cnop M Diabetologia 53, 321-330, 2010).

As noted by the reviewer, like prior groups, we removed lipofuscin as a confounding variable. We used upper and lower photon count thresholds to remove lipofuscin signals with extreme high intensities. We now clarified this in the methods section. In the new Supplementary Figure 4 we show the autofluorescence pattern in mouse and human islets in NAD(P)H channel (440-500nm) and the lipofuscin bodies shown in the lipofuscin Channel (560- 620nm). In both human and mouse islet studies, we used the upper threshold of photon counts to eliminate lipofuscin pixels with extremely high intensity in NAD(P)H channels (Supplementary Table 3).

In mouse islets, we analyzed the average bound/free NAD(P)H ratio when applying different thresholds (Supplementary Table 3) and reassuringly we do not find a difference in ratio to three decimal places between thresholds: the change in ratios in cells exposed to an increase in glucose remain consistent. For human islets, we analyzed the phasor center of lipofuscin channels and we also see no differences in the bound/free NAD(P)H ratio before and after glucose.

5) How were the “desirable” z planes selected and how were images of samples registered before and after treatment? It would also be helpful to show immunostaining that guided segmentation.

Response. The Z planes in islets were chosen to be between 10-30 μ M from the attached bottom of the islets (now added to methods). As requested we have added an example of immunostaining used to guide segmentation (Supplementary Figure 3). Having established the very different FLIM signatures in beta and alpha cells in islets where we successfully secured subsequent immunostaining of the same plane of section. We were then able to identify these cell types based on FLIM signatures, cell size (alpha cells smaller) and in mouse islets alpha cells being mostly peripheral versus beta cells centrally located.

6) There are some potential concerns about statistical analysis. Assessments involving 16 cells seems somewhat low. How many independent biological replicates did this come from? Have the authors considered donor-donor or culture-culture variability? T-tests may not be appropriate.

Response. Thanks. 8 dispersed α cells were analyzed from 2 WT mice and 8 dispersed β cells were analyzed from 1 WT mouse. 11 WT intact islets were analyzed from 7 mice and 10 hTG intact islets from 5 mice; 1-3 islets / mouse. Mice characteristics are listed in Supplementary Table 1.

Point re T-tests well taken and all comparisons now made using 1) the Mann Whitney test in WT-hTG and alpha-beta comparison, 2) paired Wilcoxon test in BG-HG comparison, as stated in new figure legend. While some p values have changed none of the comparisons the comparisons that were significantly different remain so.

7) The discussion is somewhat speculative. The manuscript would be strengthened if there were supporting measurements of ADP/ATP or membrane potential for example.

Response. While we agree with the reviewer that in future studies it would be interesting to perform subcellular ATP measurements, we respectfully propose these studies are beyond the scope of this particular manuscript, particularly given the constraints of establishing new experimental approaches with the constraints of possible laboratory work in the current environment. Rather, with the limited experimental time available we have chosen to consolidate the findings with regard to mitochondria to address the requested additional studies in the reviewers' point 8 below.

8) The authors reference the mitochondrial network multiple times, but some of their autofluorescence signal also is coming from NAD(P)H in the cytosol. This is important to consider when discussing intracellular heterogeneity and the qualitative differences they observe. Is it possible that some differences may not be related to the mitochondria?

Response. The reviewer makes a good point. The elegant studies of the Piston group note that NADPH but not NADH is generated by pyruvate in the absence of glucose stimulation, and note that this NADPH can be generated in mitochondria or cytosol by isocitrate dehydrogenase (Rocheleau JV JBC 2004). To the extent that glucose induced increases in NAD(P)H is attributable to metabolism of pyruvate by pyruvate carboxylase rather than PDH, and that the increment in NADPH was generated by cytoplasmic rather than mitochondrial isocitrate dehydrogenase, then the increased NAD(P)H pixels may be cytosolic rather than mitochondrial. We now state the assumption that subcellular regions of high bound/free NAD(P)H are located in mitochondria, but note the exception above citing the Piston paper. In order to test this assumption in our system, we performed additional studies in INS1-E cells evaluating the FLIM signature (Supplementary Figures 1A and 1B) and then the location of the mitochondrial network by IF for Tom20 (Supplementary Figure 1C). The filamentous pattern and location of the high bound/free NAD(P)H signal appears to be broadly in agreement with the Tom20 IF.

9) Minor:

a. Why is G described as a function of τ in the plots? This is unusual.

Response. Thanks, this good point comes up with other reviewers. As G is a vector without units we agree and now report G in the conventional manner for prior FLIM studies.

b. Is nucleus included in all measurements? Has very different lifetime.

Response. No, nuclei were not included as they have a very low intensity that did not meet the set minimal threshold for inclusion. We now add this point to the methods.

c. The authors write “NADH generated by oxidative phosphorylation is bound to protein complexes and has a longer lifetime”. This does not make sense because OX PHOS generally generates NAD⁺, not NADH. The bound state is not “generated” by OX PHOS as they indicate in another location in the manuscript. The bound state is generated when NADH binds to a protein complex, and this is not necessarily specific to Complex I.

Response. Good point and corrected, generated by the TCA cycle and predominantly bound to complex 1.

d. The authors mention confocal microscopy in a couple locations. If they actually performed confocal microscopy (involving use of a confocal pinhole) and did not just use the laser scanning capabilities of the SP8 for TPEF, please clarify why and what the pinhole was set to.

Response. Yes, we did perform confocal microscopy using a pinhole setting of 1 as now added to methods for fluorescent staining imaging. This pinhole setting enhanced lateral resolution and reduced out-of-focus signal in deep tissue.

e. There is a typo in the statistics section ($p < 0.5$).

Response. Thanks, corrected.

f. What was the integration time for FLIM?

The frame rate of FLIM is 0.192/s. We acquire 4-6 frame repetition until max 100 photons/pixel to reach desired resolution and photon counts for FLIM analysis. The average integration time for each frame is 26s.

Reviewer #3 (Remarks to the Author):

Pancreatic Islets and Autofluorescence based α - β separation have piqued interest in diabetic research for nearly four decades (Van de Winkel et al. 1982). Plenty of new information on islet-cell oscillations, calcium stimulus-response, and other studies have made critical advances with the help of autofluorescence-based imaging.

In this work, Wang et al. describe an autofluorescence based Glucose-FLIM assay to study metabolic responses in the pancreatic cell line(INS-1E). The authors demonstrate glucose sensitivity of the NAD(P)H FLIM method for islet cells and the ability to distinguish α and β cells using the relative change in the g-coordinate of the phasor plot. The method is further used to study the islets of transgenic diabetic mouse model(IAPP) and human islets from diabetic and non-diabetic donors. Although the work does a wonderful job in concisely present a multi-model imaging scheme for pancreatic research, I find the technical part lacks clarity in many instances.

Response. Thank you for your careful reading and evaluation of our manuscript and helpful critiques.

My comments are listed below.

* It is not clear which experiments were complemented with confocal/staining to identify α / β cells. If it was used only for dispersed single cells, how were the mouse-islets separated?

Response. Good point. As per reviewer 2's request (point 5) we provide representative immunostaining (Supplementary Figure 3) to show how we segmented beta versus alpha cells.

* Authors mention the "metabolic trajectory"; but never shows a 2D phasor plot with the metabolic trajectory. For phasor analysis experts, it would be reassuring to demonstrate the phasor trajectory as well. Only a representative image was shown in figure 1. For this representative plot, the figure does not address how the endpoints of the color-trajectory were derived. Was this experimental?

Response. Thanks again for this good point, also raised by reviewer 2 and addressed. 1) The upper and lower bounds of the color map was determined by the combined phasor cloud boundary in each set of experiments. The boundaries were different in every figure, but consistent between the panels of each figure. We added the original phasor plot to the Supplementary Figure 2) Figure 1C is the experimental overlaid phasor plot of Figure 1A and 1B.

* Line 22: Abstract: Authors mention that the work explores "glucose metabolism with regulated hormone secretion." This is misleading, in my opinion. Consider changing this to more appropriate results.

Response. Point taken and sentence deleted.

* Line 22: Abstract: "In healthy cells, glucose enhanced": consider changing "glucose"-> "addition of glucose or Stimulus of glucose or concentration of glucose or similar."

Response. Again, we agree and changed to "before and after glucose stimulation"

* The notation $G(\tau)$ for frequency domain / Fourier coefficients is confusing. Commonly used/accepted terms are $g(\omega)$ or g . $G(\tau)$ is conventionally used for lag time characteristics like FCS.

Response. We agree, and have changed the notations to that most commonly used for FLIM as suggested.

*Why was the "primary islet cells reveal differential response" study done for cell culture instead of islets? Islet staining/confocal could have been a more convincing case for the following section on islet cell separation based on morphology and FLIM (section: "glucose metabolism in intact mouse islets"). Are the dispersed islets challenging to handle on an ibidi-marked petri-dish? Other than experimental difficulties, is there another reason to not evaluate on islets instead of single cells?

Response. The advantage of dispersed primary cells over that over of cells within intact islets is that we are able to obtain greater subcellular resolution, partly because we are not imaging through a group of cells and partly because the individual cells tend to adopt a more flattened shape as they adhere to the underlying matrix.

None the less, in response to this suggestion, we have also now evaluated beta and alpha cells within intact islets as shown in the revised Figure 2G (segmentation approach is illustrated in Supplementary Figure 3). Reassuringly the pattern of FLIM signature at basal and high glucose is comparable to that in the dispersed cells.

* Line 105: "In contrast, lifetime signals indicate a predominance of glycolysis in the cell nucleus at both low and high glucose." This is hard to read from the figure panels 1d,1e. Without either a quantification of the nucleus signals or a reasonable link to the manuscript, it would be better not to comment on metabolism in the nucleus in the results section. This seems out of the scope of this work.

Response. We agree with the reviewer and have removed this as suggested, thank you.

* Fig 1 Panel f): Please provide a representative phasor in Figure 1 for LG-HG condition. This is a critical plot to validate phasors.

Response. As requested we now provide the representative phasor plot in the revised Figure 1 and Supplementary Figure 2C. As requested, Figure 1C is a representative phasor of LG-HG condition.

* I am not convinced by the use of ΔG as a "relative oxphos level." This scale is not necessarily linear for FLIM(lifetimes) or metabolism and can create systematic errors in interpretations. I have seen a similar approach in another work by Kumar et al,2020: <https://doi.org/10.1128/mBio.02730-18>. However, a direct correlation of relative respiration in cells and Δg is still to be proven with a complementary technique. Please supplement the use with published reports or flux experiments.

Response. Thanks for your advice, we upgraded the algorithm. We established the NADH metabolic trajectory by using linear regression analysis of all median G and S values. Then we mapped each data point to the regression to calculate the Bound NADH percentage in each data set. Then we used the

Bound NADH percentage and percentage change to replace ΔG to evaluate relative OXPPOS level. We agree the work by Kumar et al. has limitations by comparing the stimulated changes between different models. In contrast, we used bound/free NADH ratio for quantitative analysis. Metabolic flux studies were beyond the scope of this manuscript but by way of validation, we have added experiments in which we either selectively inhibited glycolysis by 2-deoxyglucose or oxidative phosphorylation by use of potassium cyanide (Supplementary Figure 1A and 1D respectively), and concurrently evaluated bound/free NAD(P)H in beta cells.

* Line 110: "The $G(\tau)$ score is the median lifetime vector (phasor plot center) of detected fluorescence as a fraction of 1: 1 representing fluorescence that decays quickly; and values near 0 representing very long lifetime (0 = background light)": What do authors mean by "median lifetime vector"? Is there a median calculation in the frequency domain used here? Was the kernel implemented in the phasor-space or time-space? I am also facing difficulty in understanding 1:1 relation of $\tau=0$ and $\tau=\infty$. Please explain what the scoring used for the parameter $G(\tau)$.

Responses.

- 1) The "median lifetime vector" refers to the calculation of the median of all pixels' phasor coordinates. For each sample, we take the median G and S value of all pixels as the center of the phasor plot.*
- 2) There is no median calculation in the frequency domain used.*
- 3) The kernel was implemented in phasor-space.*
- 4) On the regression line, the Free NADH percentage change scales linearly, equaling the ratio of the distance between data point and Free NADH point to the distance between data point and Bound NADH point. So Free NADH percentage, instead of $G(\tau)$ is 1:1 relation of Free NADH and Bound NADH.*

* Figure 2 Panel E: please plot the two cells separately without overlap. The clutter makes it hard to see the changes to individual cells.

Response. Thanks for your suggestion, we present β cells as dark circles and α cells as light squares for better illustration.

* Line 141: "there was an increase in oxidative phosphorylation in β cells with high glucose, extending through the mitochondrial network": Do the figures prove this claim of "extending through the mitochondrial network"? I see an increased frequency of local peaks, not a network response.

Response. Thanks for pointing out this. We softened the statements as regards pattern of the high bound ox phos signal to describe their subcellular distribution and form rather than directly ascribing these to mitochondria. That said, we have also undertaken studies of FLIM followed by immunodetection of the mitochondrial network (Supplementary Figure 1) that give some assurance that the subcellular regions of high bound/free NAD(P)H that appear in a filamentous pattern are consistent with the mitochondrial network.

* Line 208: "However, the extent of the mitochondrial network that showed suppression of oxidative

phosphorylation was greater in WT than hTG α cells ($p < 0.01$): I find the mention of mitochondrial network speculative. The figure nor the analysis does not measure the mitochondrial network extent.

Response. Thanks for your suggestion. The mitochondrial network statement has been modified to remove mitochondrial network.

* Line 433: "100 photon/pixel" What was the accumulated photon count per pixel? Are 100 photons enough for separating cells using the $G(\tau)$ parameter? Was this previously studied/reported?

Responses

(1). The photon count per pixel varies from 1-100 in each pixel. The frame repeats end when most of cytoplasm area pixels reach around 100 photons. For human islets, we allowed lipofuscin pixels to reach more than 100 photon counts to make sure get enough signal from cytoplasm area.

(2). 100 photons are enough for phasor analysis by SP8-DIVE fastFLIM filter. It will generate more than 10^5 photon count in the 256x256 imaging format for effective lifetime decay. Higher photon counts give similar phasor distribution but may cause photobleaching and phototoxicity. We found the current image setting better balanced the imaging quality and limited photobleaching.

(3). Yes, prior study "Rolf T. Borlinghaus, 2018. Lifetime – a Proper Alternative SP8 FALCON – Fluorescence lifetime images (FLIM) in an instant." Alvarez¹, Luis AJ, et al. SP8 FALCON: a novel concept in fluorescence lifetime imaging enabling video-rate confocal FLIM. (2019).

* Line 438: "FLIM data were collected in 512 x 512 pixel format at 400 Hz." : Was a 2.5ms pixel dwell time used for imaging? Is that a safe practice with 2-Photon excitation? In this context, what was the laser power used?

Response. 512x 512 400Hz was for the INS1E cell line only. The dwell time is 102.38us. We do not see laser damage in cells, and the lifetime is stable. The laser power is 9%-11% of 2.1W.

* Line 441: "FLIM was then undertaken prior to addition of glucose" -> consider replacing the verb undertaken for FLIM.

Response. Good point, changed to applied.

* Figure 5: Could authors use a color scale proportional to ΔG used to show the 3D volumes? From the presented study, it seems like ΔG is more sensitive than the phasor trajectory. In that case, a more appropriate color scale amplifying the change in g-score could increase the contrast between the samples presented in panels a and b.

Thank you for this suggestion. We addressed this by masking the range of the phasor trajectory to the region that encompasses the range of data for each experiment.

Apart from the comments listed above, I find the article exciting because of the vast amount of specimens examined and the raw effort put towards correlating metabolism using a developing science of autofluorescence-diagnostics. I appreciate the authors effort and wish the best of luck with the submission.

Response. Thank you! We have enjoyed learning about this technique and indeed enhancing that knowledge from the excellent reviews.

Reviewers' comments:

Reviewer #1 (Remarks to the Author):

The authors have made the necessary corrections.

Reviewer #2 (Remarks to the Author):

The authors have resolved almost all of the concerns from the initial round of reviews with the additional information related to their methods and results. One remaining point regarding statistical analysis remains unresolved from my view. In response to questions about the number of independent biological replicates and the potential effect of donor-donor or culture-culture variability, the authors stated the number of cells and islets analyzed. They also indicated they have changed their statistical analysis to non-parametric tests, which is fine, but doesn't address my concern about whether cells and islets can be considered independent biological replicates in their statistical analysis. Their study design has a natural hierarchy starting from independent donors or mice, to islets, to image fields, to individual cells. While donors or mice are clearly independent biological replicates, it is unclear whether all islets or cells within a group are truly independent from one another. In other words, are cells or islets from the same mouse in a given experimental group inherently more similar to each other than cells in the same group but a different mouse? Variance at all levels of their hierarchy could be considered with statistical modeling or they could perform analysis based on mean values at the hierarchy level where they can confirm they have truly independent replicates.

Reviewer #3 (Remarks to the Author):

The manuscript has improved its quality and presentation of results. I am glad to see the replies and thorough changes made. The figure changes and new metric/scoring system is all interesting. However, I have a few more minor concerns with the current version. please see the below.

Comments on current MS

- o) Fig 1: Why $G(\tau)$ is still used as a representation in Fig1?
- o) Authors include a regression line for the new results: Is the regression line fixed? or is it replotted for each distribution?
- o) Line 388: how does 100photons/pixel yield >10,000 photons/lifetime_decay_curve? (including the 5x5 median filter, it is 2500 photons). So I assume a 512x512 image was reduced to 256x256 by the Leica FLIM filter. Is this the logic here?
- o) Fig1- The red and blue marker lifetime values or (g,s) coordinate should be listed in the Figure caption
- o) Line 400: "256 x 256 pixel format at 100 Hz": Is this the frame rate? 100 frames/sec?
- o) Line 399: "FLIM was then applied prior to addition of glucose", "FLIM was then monitored at 30, 60 and 120 min": consider rephrasing the verbs: "application of FLIM" and "monitoring of FLIM".
- o) Line 413: This calculation is not clear. Which axis is the vertical projection? The intersect point to the phasor circle does not give total NAD(P)H. This is confusing. could authors provide a figure to show this calculation? Is this a projection made only on the g-axis? This description does not match the regular phasor- bound/total descriptions. : "The Bound/total NAD(P)H equals to the ratio of D-bound (distance from the vertical projected point of each data to the 100% free NAD(P)H point) to D-total (distance between two intersect points of the regression line and phasor semicircle)."
- o) Figure Suppl.2: Why do we see the second peak in the dispersed islets? Can their phasors be mapped with their representative images (if the phasors are made from one image)? so that there is proper context to the phasor location? Is the lipofuscin phasor location known?
- o) Suppl Fig 1A: could supplementary Fig1 can have color bars for panel A?
- o) Suppl Fig 1B: This is an interesting result. Is the full-circle on the phasor plot intended?

in the author's reply to response:

- o) "As requested, Figure 1C is a representative phasor of LG-HG condition.": Could both conditions

be shown in different colors? I see the one overlaid-distribution for LG-HG.

o) "We do not see laser damage in cells, and the lifetime is stableThe laser power is 9%-11% of 2.1W. ": I am surprised that 200mW laser power did no damage to live cells. What do authors mean by lifetime is stable, in the context of laser damage? Was the lifetime studied through the 102us and saw no difference? Does the Leica fast FLIM-filter let a user plot (g,s) across smaller time-bins inside a single pixel-period? I am assuming the authors meant this. But please correct me.

Referee expertise:

Referee #1: FLIM and metabolism

Referee #2: FLIM and metabolism

Referee #3: FLIM

Reviewers' comments:

Reviewer #1 (Remarks to the Author):

The authors have made the necessary corrections.

Response. Thank you.

Reviewer #2 (Remarks to the Author):

The authors have resolved almost all of the concerns from the initial round of reviews with the additional information related to their methods and results. One remaining point regarding statistical analysis remains unresolved from my view. In response to questions about the number of independent biological replicates and the potential effect of donor-donor or culture-culture variability, the authors stated the number of cells and islets analyzed. They also indicated they have changed their statistical analysis to non-parametric tests, which is fine, but doesn't address my concern about whether cells and islets can be considered independent biological replicates in their statistical analysis. Their study design has a natural hierarchy starting from independent donors or mice, to islets, to image fields, to individual cells. While donors or mice are clearly independent biological replicates, it is unclear whether all islets or cells within a group are truly independent from one another. In other words, are cells or islets from the same mouse in a given experimental group inherently more similar to each other than cells in the same group but a different mouse? Variance at all levels of their hierarchy could be considered with statistical modeling or they could perform analysis based on mean values at the hierarchy level where they can confirm they have truly independent replicates.

Response. Thank you. We apologize for missing this point in prior review and now address it as requested. The INS1 cell lines are all originally derived from a single rat insulinoma and so realistically we cannot offer biological replicates for those studies, as even though we performed multiple experiments.

To address this reasonable request from the reviewer, we now explain that both the INS-1E studies and dispersed single cell studies were the technical pilot studies to set up the method which allowed us to progress to the more important studies, given the known within islet cell to cell communication in intact isolated islets, on which the conclusions of the studies are based. We focused on the whole islets studies to address the reviewers request. We have obtained the assistance of statisticians from the Department of Medicine Statistics Core (Elashoff/Grogan) to properly assess and interpret our findings. We have now

updated the statistical methods section and results/figure legends using a linear mixed effects model with terms for random mice and/or random islets within mice or humans to account for the interdependence of replicate data.

Reviewer #3 (Remarks to the Author):

The manuscript has improved its quality and presentation of results. I am glad to see the replies and thorough changes made. The figure changes and new metric/scoring system is all interesting. However, I have a few more minor concerns with the current version. please see the below.

Response. Many thanks.

Comments on current MS

1) Fig 1: Why $G(\tau)$ is still used as a representation in Fig1?

Responses. For clarification, $G(\tau)$ was not used in any of the presented data as requested by the prior reviews. $G(\tau)$ is now only shown in panel 1c to label the x-axis as is the convention of phasor plots. As requested, all the presented data is the ratio of Bound/total NADH.

2) Authors include a regression line for the new results: Is the regression line fixed? or is it replotted for each distribution?

Response. Thank you for the question. The separate regression is created for each biological group data set (i.e. INS-1E, dispersed islet cells, whole mouse islets and whole human islets) as follows (illustrated diagrammatically below:

Step 1: Phasor FLIM distributions (G, S coordinates) were derived from a Fourier transform of lifetime (τ) data (Digman et al., 2008), with each pixel in the image corresponding to a point in the phasor plot histogram.

*Step 2: We applied an intensity threshold and a region of interest filter to remove background pixels outside of the cells. We then calculated the **median** of the cloud of points on the phasor histogram.*

Step 3: The above medians of the G and S coordinates were plotted on a phasor plot for each cell in each experimental condition (as an example, hypothetical medians for five cells at the same experimental conditions are shown as points in different shades of green in the diagram below).

Step 4: The $\tau=0.4$ ns (τ_1) was plotted on the phasor plot to reflect the position of 100% free NAD(P)H (blue diamond).

Step 5: A linear regression line, passing through the fixed point τ_1 , calculated using all experimental data points from a specific biological data set was plotted. The other intersection of the regression line on the universal circle (τ_2 ; red triangle in the diagram below) represents the average Bound-NAD(P)H lifetime for the data set.

Step 6: The ratio of bound/total NAD(P)H was computed from the projections of the data points onto the regression line, with Tau1 representing 0 and Tau2 representing 1.

Diagram to illustrate steps 1-6 above

3) Line 388: how does 100photons/pixel yield >10,000 photons/lifetime decay curve? (including the 5x5 median filter, it is 2500 photons). So I assume a 512x512 image was reduced to 256x256 by the Leica FLIM filter. Is this the logic here?

Response. Sorry, we were not clear. We set the collection conditions to collect 100 photons/pixel in the cell cytoplasm for either 256 x256 or 512 x 512 format images. The median filter was applied during imaging processing, following the published phasor approach [Ranjit, Suman. 2018. Fit-free analysis of fluorescence lifetime imaging data using the phasor approach. Nature Protocols. 13. 10.1038/s41596-018-0026-5].

Since the cytoplasm reflects ~10% of the pixels, this means we are collecting more than 200,000 (600,000 for 512x512) photons per image. We have clarified the text as follows in line 397-399:

*“Excitation intensity and number of frames collected were adjusted to collect >100 photons/pixel in the cytoplasm; thus, more than 10⁵ photon counts contributed to the phasor histograms in **each of the** optical sections.”*

4) Fig1- The red and blue marker lifetime values or (g,s) coordinate should be listed in the Figure caption

Response. As requested we have added the lifetime values in the figure 1 legend.

5) Line 400: "256 × 256 pixel format at 100 Hz": Is this the frame rate? 100 frames/sec?

Response. Thank you for pointing out this confusion. No, 100Hz is not the frame rate. 100 Hz is the line rate. 100Hz is the scan speed as 100 lines/sec. We rephrased the sentence as: "...at 0.4 frames per sec." in new line 411.

6) Line 399: "FLIM was then applied prior to addition of glucose", "FLIM was then monitored at 30, 60 and 120 min": consider rephrasing the verbs: "application of FLIM" and "monitoring of FLIM".

Response. Thank you for the suggestions. The text was changed in new line 408 as proposed.

"We evaluated the islets by FLIM prior to the addition of glucose to the media to reach a concentration of 16 mM glucose and then evaluated the islets by FLIM at 30, 60 and 120 min."

7) Line 413: This calculation is not clear. Which axis is the vertical projection? The intersect point to the phasor circle does not give total NAD(P)H. This is confusing. could authors provide a figure to show this calculation? Is this a projection made only on the g-axis? This description does not match the regular phasor- bound/total descriptions. : "The Bound/total NAD(P)H equals to the ratio of D-bound (distance from the vertical projected point of each data to the 100% free NAD(P)H point) to D-total (distance between two intersect points of the regression line and phasor semicircle)."

Response. Thank you for point out the confusion in this description (this relates to point 2, above).

1)The projection is from the data point to the regression line, not the vertical projection as mistakenly stated, as shown in the diagram above.

2) The regression line (black dash line) was calculated using all the experimental points from each biological data set, passing through the fixed point of free NAD(P)H (blue diamond, tau1) to. The other intersection of the regression line with the universal circle in the phasor plot corresponds to the mean bound NAD(P)H lifetime of the data set (red triangle, tau 2). The mean of the Bound/total NAD(P)H ratio is presented in Supplementary Table 4 and as shown as the black close circle in the diagram below.

Given points 2 and 7, we have rewritten the description of the calculation as follows;

"For each data set, the Bound/total NAD(P)H value was determined using a linear regression line fit to the G and S for each cell in the set, with the fixed point of 100% free NAD(P)H (tau 1). The other intersection of the regression line with the universal circle of the phasor plot (tau 2) represents the mean lifetime of 100% Bound NAD(P) for that data set. The Bound/total NAD(P)H for each data point was calculated by projecting each GS value onto the regression line, and then related to the distance between tau 1 and tau 2. The mean of the Bound/total NAD(P)H ratio is presented in Supplementary Table 4. Δ Bound/total NAD(P)H is defined as the Bound/total NAD(P)H after treatment minus Bound/total NAD(P)H before treatment."

8) Figure Suppl.2: Why do we see the second peak in the dispersed islets? Can their phasors be mapped with their representative images (if the phasors are made from one image)? so that there is proper context to the phasor location? Is the lipofuscin phasor location known?

Response. Good point.

The second peak was the background signal, which was included in error. Thank you, we have now corrected the panel. The previous panel, with the pixels in the image highlighted that contributed the second peak to the phasor plot, is shown below for reference.

Supplementary Figure 2

Previous

Phasor plot of data set with background optical section

New

FLIM map of background optical section

In our previous analyses, we eliminated the lipofuscin peak by gating out the high intensity pixels (Supplementary Figure 4). To show the position of lipofuscin on the phasor plot, we show the phasor position of these intense pixels below.

a.

Beta cell

b.

Phasor plot of the beta cell

c.

Threshold Phasor plot of lipofuscin

Phasor plot of lipofuscin signal in NAD(P)H channel (440-500 nm). a. High intensity pixels of lipofuscin in a beta cell were highlighted in magenta. b. Phasor plot of the beta cell. Data points corresponding to lipofuscin pixels were circled in magenta. c. The lipofuscin peak by gating in the high intensity pixels.

9) Suppl Fig 1A: could supplementary Fig1 can have color bars for panel A?

Response. As requested, a color bar was added to Suppl Fig 1A.

10) Suppl Fig 1B: This is an interesting result. Is the full-circle on the phasor plot intended?

Response. Good pickup. The full-circle was not intended, but is a mistake from an incorrect figure format conversion. Now corrected, thank you.

in the author's reply to response:

11) "As requested, Figure 1C is a representative phasor of LG-HG condition.": Could both conditions be shown in different colors? I see the one overlaid-distribution for LG-HG.

Response. As requested we have created a figure that overlays the data for the two conditions in two different colors so they can be compared.

12) "We do not see laser damage in cells, and the lifetime is stable. The laser power is 9%-11% of 2.1W.": I am surprised that 200mW laser power did no damage to live cells. What do authors mean by lifetime is stable, in the context of laser damage? Was the lifetime studied through the 102us and saw no difference? Does the Leica fast FLIM-filter let a user plot (g,s) across smaller time-bins inside a single pixel-period? I am assuming the authors meant this. But please correct me.

Responses. Thank you for the opportunity to clarify, as we might have implied the use of a far larger laser power through the objective lens than was actually the case. The power out the objective lens is far below the 200mW implied in our quote. The actual value out the objective lens at 10% power was 800uW. Line 392 was changed as: "... 2- photon excitation (Spectra-Physics InSight X3 tunable laser, 0.8mW average power)"

In setting up these studies, we performed pilot studies to optimize the FLIM measurements, exploring different culture conditions and laser exposure. The studies reported here were then performed using laser exposure that was well below the levels at which we had observed damage of the cells in extended imaging (loss of cell morphology, loss of cell adherence, or fluctuations in measured fluorescence lifetimes).

In the studies reported here, the NAD(P)H lifetimes were stable through repeated image acquisitions in the same glucose levels. We imaged both INSE1 cells and islets twice at the same basal glucose, separated by 30 min, and obtained the same NAD(P)H lifetime between the 2 data acquisitions.

The pixel dwell time we used in our imaging was 102 μ s, and the Leica fast FLIM-filter is not able to plot (g,s) across smaller time-bins than the full dwell times. Our comment about stable lifetimes was between sequential frames, or between multiple frame images separated by 30 minutes.

We have clarified the nature of the stability to the methods section to address the reviewer's important question about photo-damage to the samples.

Dear Dr. Butler,

Your manuscript entitled "Live cell imaging of glucose-induced metabolic coupling of β and α cell metabolism in health and type 2 diabetes" has now been seen by 3 referees. While reviewers are mostly happy with the revised manuscript, reviewer #2 still has some concern about the statistical analysis done in this study. We are still interested in the possibility of publishing your study in Communications Biology, but would like to consider your response to this remaining concern and the minor comments of reviewer #3 in the form of a revised manuscript before we make a final decision on publication.

We therefore invite you to revise and resubmit your manuscript, taking into account the points raised. Please highlight all changes in the manuscript text file.

We are committed to providing a fair and constructive peer-review process. Do not hesitate to contact us if you wish to discuss the revision in more detail or if there are specific requests from the reviewers that you believe are technically impossible or unlikely to yield a meaningful outcome.

Please use the following link to submit your revised manuscript, point-by-point response to the referees' comments (which should be in a separate document to the cover letter) and any additional files:

<https://mts-commsbio.nature.com/cgi-bin/main.plex?el=A2Cx2BpX6B6bhh1I6A9ftd7W45qFyfWMp3y9gHXJBaJQZ>

We would expect revisions of this nature to take around three months, but appreciate that every situation is unique. We look forward to receiving your revised manuscript when it is ready, and will not enforce a hard deadline on this revision.

Please do not hesitate to contact me if you have any questions or would like to discuss these revisions further. We look forward to seeing the revised manuscript and thank you for the opportunity to review your work.

Best regards,

Jung-Eun Lee, PhD
Senior Editor, Communications Biology
One New York Plaza, Suite 4600
New York, NY 10004-1562
orcid.org/0000-0003-0184-3440
jung-eun.lee@nature.com

REVIEWERS' COMMENTS:

Reviewer #2 (Remarks to the Author):

The reviewers have addressed all concerns.